# Understanding and Improving Information Transfer in Multi-Task Learning

**Sen Wu**[*]
Stanford University

**Hongyang R. Zhang**[*]
University of Pennsylvania

**Christopher Ré**
Stanford University

## Abstract

We investigate multi-task learning approaches that use a shared feature representation for all tasks. To better understand the transfer of task information, we study an architecture with a shared module for all tasks and a separate output module for each task. We study the theory of this setting on linear and ReLU-activated models. Our key observation is that whether or not tasks' data are well-aligned can significantly affect the performance of multi-task learning. We show that misalignment between task data can cause negative transfer (or hurt performance) and provide sufficient conditions for positive transfer. Inspired by the theoretical insights, we show that aligning tasks' embedding layers leads to performance gains for multi-task training and transfer learning on the GLUE benchmark and sentiment analysis tasks; for example, we obtain a 2.35% GLUE score average improvement on 5 GLUE tasks over BERT$_{\text{LARGE}}$ using our alignment method. We also design an SVD-based task reweighting scheme and show that it improves the robustness of multi-task training on a multi-label image dataset.

## 1 Introduction

Multi-task learning has recently emerged as a powerful paradigm in deep learning to obtain language (Devlin et al. (2018); Liu et al. (2019a;b)) and visual representations (Kokkinos (2017)) from large-scale data. By leveraging supervised data from related tasks, multi-task learning approaches reduce the expensive cost of curating the massive per-task training data sets needed by deep learning methods and provide a shared representation which is also more efficient for learning over multiple tasks. While in some cases, great improvements have been reported compared to single-task learning (McCann et al. (2018)), practitioners have also observed problematic outcomes, where the performances of certain tasks have decreased due to task interference (Alonso and Plank (2016); Bingel and Søgaard (2017)). Predicting when and for which tasks this occurs is a challenge exacerbated by the lack of analytic tools. In this work, we investigate key components to determine whether tasks interfere *constructively* or *destructively* from theoretical and empirical perspectives. Based on these insights, we develop methods to improve the effectiveness and robustness of multi-task training.

There has been a large body of algorithmic and theoretical studies for kernel-based multi-task learning, but less is known for neural networks. The conceptual message from the earlier work (Baxter (2000); Evgeniou and Pontil (2004); Micchelli and Pontil (2005); Xue et al. (2007)) show that multi-task learning is effective over "similar" tasks, where the notion of similarity is based on the single-task models (e.g. decision boundaries are close). The work on structural correspondence learning (Ando and Zhang (2005); Blitzer et al. (2006)) uses alternating minimization to learn a shared parameter and separate task parameters. Zhang and Yeung (2014) use a parameter vector for each task and learn task relationships via $l_2$ regularization, which implicitly controls the capacity of the model. These results are difficult to apply to neural networks: it is unclear how to reason about neural networks whose feature space is given by layer-wise embeddings.

To determine whether two tasks interfere constructively or destructively, we investigate an architecture with a shared module for all tasks and a separate output module for each task (Ruder (2017)). See Figure 1 for an illustration. Our motivating observation is that in addition to model similarity which affects the type of interference, task data similarity plays a second-order effect after controlling model similarity. To illustrate the idea, we consider three tasks with the same number of data

---

[*]Equal contribution. Correspondence to {senwu,hongyang,chrismre}@cs.stanford.edu

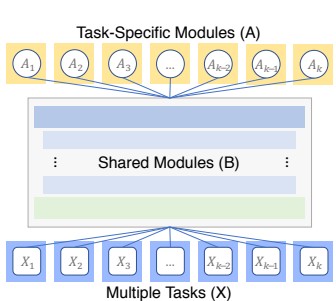

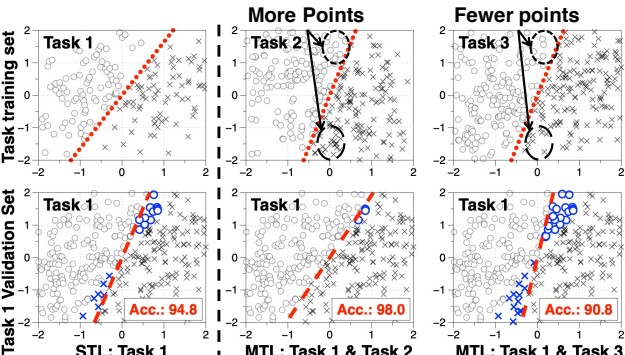

Figure 1: An illustration of the multi-task learning architecture with a shared lower module $B$ and $k$ task-specific modules $\{A_i\}_{i=1}^{k}$.

Figure 2: Positive vs. Negative transfer is affected by the data – not just the model. See lower right-vs-mid. Task 2 and 3 have the same model (dotted lines) but different data distributions. Notice the difference of data in circled areas.

samples where task 2 and 3 have the same decision boundary but different data distributions (see Figure 2 for an illustration). We observe that training task 1 with task 2 or task 3 can either improve or hurt task 1's performance, depending on the amount of contributing data along the decision boundary! This observation shows that by measuring the similarities of the task data and the models separately, we can analyze the interference of tasks and attribute the cause more precisely.

Motivated by the above observation, we study the theory of multi-task learning through the shared module in linear and ReLU-activated settings. Our theoretical contribution involves three components: the *capacity of the shared module*, *task covariance*, and the *per-task weight of the training procedure*. The capacity plays a fundamental role because, if the shared module's capacity is too large, there is no interference between tasks; if it is too small, there can be destructive interference. Then, we show how to determine interference by proposing a more fine-grained notion called *task covariance* which can be used to measure the alignment of task data. By varying task covariances, we observe both positive and negative transfers from one task to another! We then provide sufficient conditions which guarantee that one task can transfer positively to another task, provided with sufficiently many data points from the contributor task. Finally, we study how to assign per-task weights for settings where different tasks share the same data but have different labels.

**Experimental results.** Our theory leads to the design of two algorithms with practical interest. First, we propose to align the covariances of the task embedding layers and present empirical evaluations on well-known benchmarks and tasks. On 5 tasks from the General Language Understanding Evaluation (GLUE) benchmark (Wang et al. (2018b)) trained with the BERT_LARGE model by Devlin et al. (2018), our method improves the result of BERT_LARGE by a 2.35% average GLUE score, which is the standard metric for the benchmark. Further, we show that our method is applicable to transfer learning settings; we observe up to 2.5% higher accuracy by transferring between six sentiment analysis tasks using the LSTM model of Lei et al. (2018).

Second, we propose an SVD-based task reweighting scheme to improve multi-task training for settings where different tasks have the same features but different labels. On the ChestX-ray14 dataset, we compare our method to the unweighted scheme and observe an improvement of 0.4% AUC score on average for all tasks . In conclusion, these evaluations confirm that our theoretical insights are applicable to a broad range of settings and applications.

## 2 THREE COMPONENTS OF MULTI-TASK LEARNING

We study multi-task learning (MTL) models with a shared module for all tasks and a separate output module for each task. We ask: What are the key components to determine whether or not MTL is better than single-task learning (STL)? In response, we identify three components: *model capacity*, *task covariance*, and *optimization scheme*. After setting up the model, we briefly describe the role of model capacity. We then introduce the notion of *task covariance*, which comprises the bulk of the section. We finish by showing the implications of our results for choosing optimization schemes.

## 2.1 Modeling Setup

We are given $k$ tasks. Let $m_i$ denote the number of data samples of task $i$. For task $i$, let $X_i \in \mathbb{R}^{m_i \times d}$ denote its covariates and let $y_i \in \mathbb{R}^{m_i}$ denote its labels, where $d$ is the dimension of the data. We have assumed that all the tasks have the same input dimension $d$. This is not a restrictive assumption and is typically satisfied, e.g. for word embeddings on BERT, or by padding zeros to the input otherwise. Our model assumes the output label is 1-dimensional. We can also model a multi-label problem with $k$ types of labels by having $k$ tasks with the same covariates but different labels. We consider an MTL model with a shared module $B \in \mathbb{R}^{d \times r}$ and a separate output module $A_i \in \mathbb{R}^r$ for task $i$, where $r$ denotes the output dimension of $B$. See Figure 1 for the illustration. We define the objective of finding an MTL model as minimizing the following equation over $B$ and the $A_i$'s:

$$f(A_1, A_2, \ldots, A_k; B) = \sum_{i=1}^k L\left(g(X_i B)A_i, y_i\right), \tag{1}$$

where $L$ is a loss function such as the squared loss. The activation function $g : \mathbb{R} \to \mathbb{R}$ is applied on every entry of $X_i B$. In equation 1, all data samples contribute equally. Because of the differences between tasks such as data size, it is natural to re-weight tasks during training:

$$f(A_1, A_2, \ldots, A_k; B) = \sum_{i=1}^k \alpha_i \cdot L(g(X_i B)A_i, y_i), \tag{2}$$

This setup is an abstraction of the hard parameter sharing architecture (Ruder (2017)). The shared module $B$ provides a universal representation (e.g., an LSTM for encoding sentences) for all tasks. Each task-specific module $A_i$ is optimized for its output. We focus on two models as follows.

*The single-task linear model.* The labels $y$ of each task follow a linear model with parameter $\theta \in \mathbb{R}^d$: $y = X\theta + \varepsilon$. Every entry of $\varepsilon$ follows the normal distribution $\mathcal{N}(0, \sigma^2)$ with variance $\sigma^2$. The function $g(XB) = XB$. This is a well-studied setting for linear regression (Hastie et al. (2005)).

*The single-task ReLU model.* Denote by $\text{ReLU}(x) = \max(x, 0)$ for any $x \in \mathbb{R}$. We will also consider a non-linear model where $X\theta$ goes through the ReLU activation function with $a \in \mathbb{R}$ and $\theta \in \mathbb{R}^d$: $y = a \cdot \text{ReLU}(X\theta) + \varepsilon$, which applies the ReLU activation on $X\theta$ entrywise. The encoding function $g(XB)$ then maps to $\text{ReLU}(XB)$.

**Positive vs. negative transfer.** For a source task and a target task, we say the source task transfers *positively* to the target task, if training both through equation 1 improves over just training the target task (measured on its validation set). *Negative* transfer is the converse of positive transfer.

**Problem statement.** Our goal is to analyze the three components to determine positive vs. negative transfer between tasks: model capacity ($r$), task covariances ($\{X_i^\top X_i\}_{i=1}^k$) and the per-task weights ($\{\alpha_i\}_{i=1}^k$). We focus on regression tasks under the squared loss but we also provide synthetic experiments on classification tasks to validate our theory.

**Notations.** For a matrix $X$, its column span is the set of all linear combinations of the column vectors of $X$. Let $X^\dagger$ denote its pseudoinverse. Given $u, v \in \mathbb{R}^d$, $\cos(u, v)$ is equal to $u^\top v / (\|u\| \cdot \|v\|)$.

## 2.2 Model Capacity

We begin by revisiting the role of model capacity, i.e. the output dimension of $B$ (denoted by $r$). We show that as a rule of thumb, $r$ should be smaller than the sum of capacities of the STL modules.

**Example.** Suppose we have $k$ linear regression tasks using the squared loss, equation 1 becomes:

$$f(A_1, A_2, \ldots, A_k; B) = \sum_{i=1}^k \|X_i B A_i - y_i\|_F^2. \tag{3}$$

The optimal solution of equation 3 for task $i$ is $\theta_i = (X_i^\top X_i)^\dagger X_i^\top y_i \in \mathbb{R}^d$. Hence a capacity of 1 suffices for each task. We show that if $r \geq k$, then there is no transfer between any two tasks.

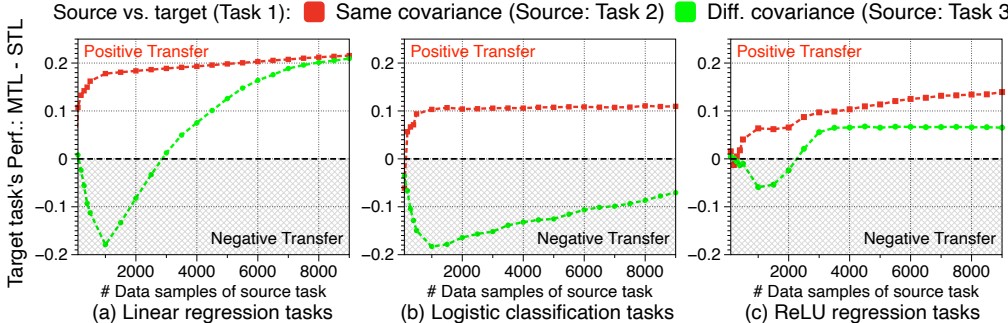

Figure 3: Performance improvement of a target task (Task 1) by MTL with a source task vs. STL. Red: positive transfer when the source is Task 2, which has the same covariance matrix with target. Green: negative (to positive) transfer when the source is Task 3, which has a different covariance from the target, as its # of samples increases. See the example below for the definition of each task.

**Proposition 1.** *Let $r \geq k$. There exists an optimum $B^\star$ and $\{A_i^\star\}_{i=1}^k$ of equation 3 where $B^\star A_i^\star = \theta_i$, for all $i = 1, 2, \ldots, k$.*

To illustrate the idea, as long as $B^\star$ contains $\{\theta_i\}_{i=1}^k$ in its column span, there exists $A_i^\star$ such that $B^\star A_i^\star = \theta_i$, which is optimal for equation 3 with minimum error. But this means no transfer among any two tasks. This can hurt generalization if a task has limited data, in which case its STL solution overfits training data, whereas the MTL solution can leverage other tasks' data to improve generalization. The proof of Proposition 1 and its extension to ReLU settings are in Appendix A.1.

**Algorithmic consequence.** The implication is that limiting the shared module's capacity is necessary to enforce information transfer. If the shared module is too small, then tasks may interfere negatively with each other. But if it is too large, then there may be no transfer between tasks. In Section 3.3, we verify the need to carefully choose model capacity on a wide range of neural networks including CNN, LSTM and multi-layer perceptron.

## 2.3 TASK COVARIANCE

To show how to quantify task data similarity, we illustrate with two regression tasks under the linear model without noise: $y_1 = X_1\theta_1$ and $y_2 = X_2\theta_2$. By Section 2.2, it is necessary to limit the capacity of the shared module to enforce information transfer. Therefore, we consider the case of $r = 1$. Hence, the shared module $B$ is now a $d$-dimensional vector, and $A_1, A_2$ are both scalars.

A natural requirement of task similarity is for the STL models to be similar, i.e. $|\cos(\theta_1, \theta_2)|$ to be large. To see this, the optimal STL model for task 1 is $(X_1^\top X_1)^{-1} X_1^\top y_1 = \theta_1$. Hence if $|\cos(\theta_1, \theta_2)|$ is 1, then tasks 1 and 2 can share a model $B \in \mathbb{R}^d$ which is either $\theta_1$ or $-\theta_1$. The scalar $A_1$ and $A_2$ can then transform $B$ to be equal to $\theta_1$ and $\theta_2$.

Is this requirement sufficient? Recall that in equation 3, the task data $X_1$ and $X_2$ are both multiplied by $B$. If they are poorly "aligned" geometrically, the performance could suffer. How do we formalize the geometry between task alignment? In the following, we show that the covariance matrices of $X_1$ and $X_2$, which we define to be $X_1^\top X_1$ and $X_2^\top X_2$, captures the geometry. We fix $|\cos(\theta_1, \theta_2)|$ to be close to 1 to examine the effects of task covariances. In Appendix A.2.1 we fix task covariances to examine the effects of model cosine similarity. Concretely, equation 3 reduces to:

$$\max_{B \in \mathbb{R}^d} h(B) = \langle \frac{X_1 B}{\|X_1 B\|}, y_1 \rangle^2 + \langle \frac{X_2 B}{\|X_2 B\|}, y_2 \rangle^2, \tag{4}$$

where we apply the first-order optimality condition on $A_1$ and $A_2$ and simplify the equation. Specifically, we focus on a scenario where task 1 is the *source* and task 2 is the *target*. *Our goal is to determine when the source transfers to the target positively or negatively in MTL.* Determining the type of transfer from task 2 to task 1 can be done similarly. Answering the question boils down to studying the angle or cosine similarity between the optimum of equation 4 and $\theta_2$.

**Example.** In Figure 3, we show that by varying task covariances and the number of samples, we can observe both positive and negative transfers. The conceptual message is the same as Figure 2; we

---

**Algorithm 1** Covariance alignment for multi-task training

---

**Require:** Task embedding layers $X_1 \in \mathbb{R}^{m_1 \times d}, X_2 \in \mathbb{R}^{m_2 \times d}, \ldots, X_k \in \mathbb{R}^{m_k \times d}$, shared module $B$
**Parameter:** Alignment matrices $R_1, R_2, \ldots, R_k \in \mathbb{R}^{d \times d}$ and output modules $A_1, A_2, \ldots, A_k \in \mathbb{R}^r$
  1: Let $Z_i = X_i R_i$, for $1 \leq i \leq k$.
   Consider the following modified loss (with $B$ being fixed):
   $$\hat{f}(A_1, \ldots, A_k; R_1, \ldots, R_k) = \sum_{i=1}^k L(g(Z_i B)A_i, y_i) = \sum_{i=1}^k L(g(X_i R_i B)A_i, y_i)$$
  2: Minimize $\hat{f}$ by alternatively applying a gradient descent update on $A_i$ and $R_i$, given a sampled data batch from task $i$.
   Other implementation details are described in Appendix B.3.

---

describe the data generation process in more detail. We use 3 tasks and measure the type of transfer from the source to the target. The $x$-axis is the number of data samples from the source. The $y$-axis is the target's performance improvement measured on its validation set between MTL minus STL.

*Data generation.* We have $|\cos(\theta_1, \theta_2)| \approx 1$ (say 0.96). For $i \in \{1, 2, 3\}$, let $R_i \subseteq \mathbb{R}^{m_i \times d}$ denote a random Gaussian matrix drawn from $\mathcal{N}(0, 1)$. Let $S_1, S_2 \subseteq \{1, 2, \ldots, d\}$ be two disjoint sets of size $d/10$. For $i = 1, 2$, let $D_i$ be a diagonal matrix whose entries are equal to a large value $\kappa$ (e.g. $\kappa = 100$) for coordinates in $S_i$ and 1 otherwise. Let $Q_i \subseteq \mathbb{R}^{d \times d}$ denote an orthonormal matrix, i.e. $Q_i^\top Q_i$ is equal to the identity matrix, orthogonalized from a random Gaussian matrix.

Then, we define the 3 tasks as follows. (i) Task 1 (target): $X_1 = R_1 Q_1 D_1$ and $y_1 = X_1 \theta_1$. (ii) Task 2 (source task for red line): $X_2 = R_2 Q_1 D_1$ and $y_2 = X_2 \theta_2$. (iii) Task 3 (source task for green line): $X_3 = R_3 Q_2 D_2$ and $y_3 = X_3 \theta_2$. Task 1 and 2 have the same covariance matrices but task 1 and 3 have different covariance matrices. Intuitively, the signals of task 1 and 3 lie in different subspaces, which arise from the difference in the diagonals of $D_i$ and the orthonormal matrices.

*Analysis.* Unless the source task has lots of samples to estimate $\theta_2$, which is much more than the samples needed to estimate only the coordinates of $S_1$, the effect of transferring to the target is small. We observe similar results for logistic regression tasks and for ReLU-activated regression tasks.

**Theory.** We rigorously quantify how many data points is needed to guarantee positive transfer. The folklore in MTL is that when a source task has a lot of data but the related target task has limited data, then the source can often transfer positively to the target task. Our previous example shows that by varying the source's number of samples and its covariance, we can observe both types of transfer. *How much data do we need from the source to guarantee a positive transfer to the target?* We show that this depends on the condition numbers of both tasks' covariances.

**Theorem 2** (informal). *For $i = 1, 2$, let $y_i = X_i \theta_i + \varepsilon_i$ denote two linear regression tasks with parameters $\theta_i \in \mathbb{R}^d$ and $m_i$ number of samples. Suppose that each row of the source task $X_1$ is drawn independently from a distribution with covariance $\Sigma_1 \subseteq \mathbb{R}^{d \times d}$ and bounded $l_2$-norm. Let $c = \kappa(X_2)\sin(\theta_1, \theta_2)$ and assume that $c \leq 1/3$. Denote by $(B^\star, A_1^\star, A_2^\star)$ the optimal MTL solution. With high probability, when $m_1$ is at least on the order of $(\kappa^2(\Sigma_1) \cdot \kappa^4(X_2) \cdot \|y_2\|^2)/c^4$, we have*

$$\|B^\star A_2^\star - \theta_2\|/\|\theta_2\| \leq 6c + \frac{1}{1 - 3c}\frac{\|\varepsilon_2\|}{\|X_2\theta_2\|}. \tag{5}$$

Recall that for a matrix $X$, $\kappa(X)$ denotes its condition number. Theorem 2 quantifies the trend in Figure 3, where the improvements for task 2 reaches the plateau when $m_1$ becomes large enough.

The parameter $c$ here indicates how similar the two tasks are. The smaller $\sin(\theta_1, \theta_2)$ is, the smaller $c$ is. As an example, if $\sin(\theta_1, \theta_2) \leq \delta/\kappa(X_2)$ for some $\delta$, then equation 5 is at most $O(\delta) + \|\varepsilon_2\|/\|X_2\theta_2\|$.[1] The formal statement, its proof and discussions on the assumptions are deferred to Appendix A.2.2.

**The ReLU model.** We show a similar result for the ReLU model, which requires resolving the challenge of analyzing the ReLU function. We use a geometric characterization for the ReLU function under distributional input assumptions by Du et al. (2017). The result is deferred to Appendix A.2.3.

---

[1] The estimation error of $\theta_2$ is upper bounded by task 2's signal-to-noise ratio $\|\varepsilon_2\|/\|X_2\theta_2\|$. This dependence arises because the linear component $A_2^\star$ fits the projection of $y_2$ to $X_2 B^\star$. So even if $B^\star$ is equal to $\theta_2$, there could still be an estimation error out of $A_2^\star$, which cannot be estimated from task 1's data.

---

**Algorithm 2** An SVD-based task re-weighting scheme

---

**Input:** $k$ tasks: $(X, y_i) \in (\mathbb{R}^{m \times d}, \mathbb{R}^m)$; a rank parameter $r \in \{1, 2, \ldots, k\}$
**Output:** A weight vector: $\{\alpha_1, \alpha_2, \ldots, \alpha_k\}$
  1: Let $\theta_i = X^\top y_i$.
  2: $U_r, D_r, V_r = \text{SVD}_r(\theta_1, \theta_2, \ldots, \theta_k)$, i.e. the best rank-$r$ approximation to the $\theta_i$'s.
  3: Let $\alpha_i = \|\theta_i^\top U_r\|$, for $i = 1, 2, \ldots, k$.

---

**Algorithmic consequence.** An implication of our theory is a *covariance alignment method* to improve multi-task training. For the $i$-th task, we add an alignment matrix $R_i$ before its input $X_i$ passes through the shared module $B$. Algorithm 1 shows the procedure.

We also propose a metric called *covariance similarity score* to measure the similarity between two tasks. Given $X_1 \in \mathbb{R}^{m_1 \times d}$ and $X_2 \in \mathbb{R}^{m_2 \times d}$, we measure their similarity in three steps: (a) The covariance matrix is $X_1^\top X_1$. (b) Find the best rank-$r_1$ approximation to be $U_{1,r_1} D_{1,r_1} U_{1,r_1}^\top$, where $r_1$ is chosen to contain $99\%$ of the singular values. (c) Apply step (a),(b) to $X_2$, compute the score:

$$\text{Covariance similarity score} := \frac{\|(U_{1,r_1} D_{1,r_1}^{1/2})^\top U_{2,r_2} D_{2,r_2}^{1/2}\|_F}{\|U_{1,r_1} D_{1,r_1}^{1/2}\|_F \cdot \|U_{2,r_2} D_{2,r_2}^{1/2}\|_F}. \tag{6}$$

The nice property of the score is that it is invariant to rotations of the columns of $X_1$ and $X_2$.

## 2.4 Optimization Scheme

Lastly, we consider the effect of re-weighting the tasks (or their losses in equation 2). When does re-weighting the tasks help? In this part, we show a use case for improving the robustness of multi-task training in the presence of label noise. The settings involving label noise can arise when some tasks only have weakly-supervised labels, which have been studied before in the literature (e.g. Mintz et al. (2009); Pentina and Lampert (2017)). We start by describing a motivating example.

Consider two tasks where task 1 is $y_1 = X\theta$ and task 2 is $y_2 = X\theta + \varepsilon_2$. If we train the two tasks together, the error $\varepsilon_2$ will add noise to the trained model. However, by up weighting task 1, we reduce the noise from task 2 and get better performance. To rigorously study the effect of task weights, we consider a setting where all the tasks have the same data but different labels. This setting arises for example in multi-label image tasks. We derive the optimal solution in the linear model.

**Proposition 3.** *Let the shared module have capacity $r \leq k$. Given $k$ tasks with the same covariates $X \subseteq \mathbb{R}^{m \times d}$ but different labels $\{y_i\}_{i=1}^k$. Let $X$ be full rank and $UDV^\top$ be its SVD. Let $Q_r Q_r^\top$ be the best rank-$r$ approximation to $\sum_{i=1}^k \alpha_i U^\top y_i y_i^\top U$. Let $B^\star \subseteq \mathbb{R}^{d \times r}$ be an optimal solution for the re-weighted loss. Then the column span of $B^\star$ is equal to the column span of $(X^\top X)^{-1} V D Q_r$.*

We can also extend Proposition 3 to show that all local minima of equation 3 are global minima in the linear setting. We leave the proof to Appendix A.3. We remark that this result does not extend to the non-linear ReLU setting and leave this for future work.

Based on Proposition 3, we provide a rigorous proof of the previous example. Suppose that $X$ is full rank, $(X^\top X)^\dagger X [\alpha_1 y_1, \alpha_2 y_2] = [\alpha_1 \theta, \alpha_2 \theta + \alpha_2 (X^\top X)^{-1} X \varepsilon_2]$. Hence, when we increase $\alpha_1$, $\cos(B^\star, \theta)$ increases closer to 1.

**Algorithmic consequence.** Inspired by our theory, we describe a re-weighting scheme in the presence of label noise. We compute the per-task weights by computing the SVD over $X^\top y_i$, for $1 \leq i \leq k$. The intuition is that if the label vector of a task $y_i$ is noisy, then the entropy of $y_i$ is small. Therefore, we would like to design a procedure that removes the noise. The SVD procedure does this, where the weight of a task is calculated by its projection into the principal $r$ directions. See Algorithm 2 for the description.

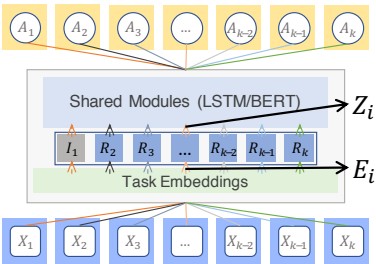

Figure 4: Illustration of the covariance alignment module on task embeddings.

# 3 EXPERIMENTS

We describe connections between our theoretical results and practical problems of interest. We show three claims on real world datasets. (i) The shared MTL module is best performing when its capacity is smaller than the total capacities of the single-task models. (ii) Our proposed covariance alignment method improves multi-task training on a variety of settings including the GLUE benchmarks and six sentiment analysis tasks. Our method can be naturally extended to transfer learning settings and we validate this as well. (iii) Our SVD-based reweighed scheme is more robust than the standard unweighted scheme on multi-label image classification tasks in the presence of label noise.

## 3.1 EXPERIMENTAL SETUP

**Datasets and models.** We describe the datasets and models we use in the experiments.

*GLUE:* GLUE is a natural language understanding dataset including question answering, sentiment analysis, text similarity and textual entailment problems. We choose BERT$_{\text{LARGE}}$ as our model, which is a 24 layer transformer network from Devlin et al. (2018). We use this dataset to evaluate how Algorithm 1 works on the state-of-the-art BERT model.

*Sentiment Analysis:* This dataset includes six tasks: movie review sentiment (MR), sentence subjectivity (SUBJ), customer reviews polarity (CR), question type (TREC), opinion polarity (MPQA), and the Stanford sentiment treebank (SST) tasks.

For each task, the goal is to categorize sentiment opinions expressed in the text. We use an embedding layer (with GloVe embeddings[2]) followed by an LSTM layer proposed by Lei et al. (2018)[3].

*ChestX-ray14:* This dataset contains 112,120 frontal-view X-ray images and each image has up to 14 diseases. This is a 14-task multi-label image classification problem. We use the CheXNet model from Rajpurkar et al. (2017), which is a 121-layer convolutional neural network on all tasks.

For all models, we share the main module across all tasks (BERT$_{\text{LARGE}}$ for GLUE, LSTM for sentiment analysis, CheXNet for ChestX-ray14) and assign a separate regression or classification layer on top of the shared module for each tasks.

**Comparison methods.** For the experiment on multi-task training, we compare Algorithm 1 by training with our method and training without it. Specifically, we apply the alignment procedure on the task embedding layers. See Figure 4 for an illustration, where $E_i$ denotes the embedding of task $i$, $R_i$ denotes its alignment module and $Z_i = E_i R_i$ is the rotated embedding.

For transfer learning, we first train an STL model on the source task by tuning its model capacity (e.g. the output dimension of the LSTM layer). Then, we fine-tune the STL model on the target task for 5-10 epochs. To apply Algorithm 1, we add an alignment module for the target task during fine-tuning.

For the experiment on reweighted schemes, we compute the per-task weights as described in Algorithm 2. Then, we reweight the loss function as in equation 2. We compare with the reweighting techniques of Kendall et al. (2018). Informally, the latter uses Gaussian likelihood to model classi-

---

[2]http://nlp.stanford.edu/data/wordvecs/glove.6B.zip
[3]We also tested with multi-layer perceptron and CNN. The results are similar (cf. Appendix B.5).

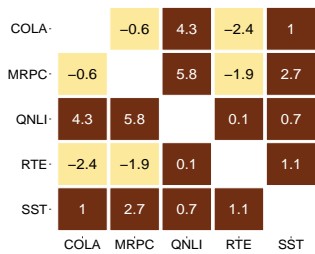

(a) MTL on GLUE over 10 task pairs

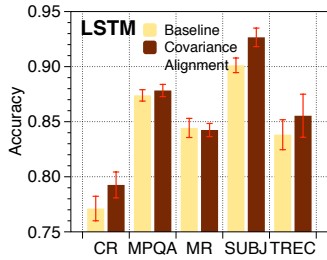

(b) Transfer learning on six sentiment analysis tasks

Figure 5: Performance improvements of Algorithm 1 by aligning task embeddings.

fication outputs. The weights, defined as inversely proportional to the variances of the Gaussian, are optimized during training. We also compare with the unweighted loss (cf. equation 1) as a baseline.

**Metric.** We measure performance on the GLUE benchmark using a standard metric called the GLUE score, which contains accuracy and correlation scores for each task.

For the sentiment analysis tasks, we measure the accuracy of predicting the sentiment opinion.

For the image classification task, we measure the area under the curve (AUC) score. We run five different random seeds to report the average results. The result of an MTL experiment is averaged over the results of all the tasks, unless specified otherwise.

For the training procedures and other details on the setup, we refer the reader to Appendix B.

## 3.2 EXPERIMENTAL RESULTS

We present use cases of our methods on open-source datasets. We expected to see improvements via our methods in multi-task and other settings, and indeed we saw such gains across a variety of tasks.

**Improving multi-task training.** We apply Algorithm 1 on five tasks (CoLA, MRPC, QNLI, RTE, SST-2) from the GLUE benchmark using a state-of-the-art language model $BERT_{LARGE}$. [4] We train the output layers $\{A_i\}$ and the alignment layers $\{R_i\}$ using our algorithm. We compare the average performance over all five tasks and find that our method outperforms $BERT_{LARGE}$ by 2.35% average GLUE score for the five tasks. For the particular setting of training two tasks, our method outperforms $BERT_{LARGE}$ on 7 of the 10 task pairs. See Figure 5a for the results.

**Improving transfer learning.** While our study has focused on multi-task learning, *transfer learning* is a naturally related goal – and we find that our method is also useful in this case. We validate this by training an LSTM on sentiment analysis. Figure 5b shows the result with SST being the source task and the rest being the target task. Algorithm 1 improves accuracy on four tasks by up to 2.5%.

**Reweighting training for the same task covariates.** We evaluate Algorithm 2 on the ChestX-ray14 dataset. This setting satisfies the assumption of Algorithm 2, which requires different tasks to have the same input data. Across all 14 tasks, we find that our reweighting method improves the technique of Kendall et al. (2018) by 0.1% AUC score. Compared to training with the unweighted loss, our method improves performance by 0.4% AUC score over all tasks.

## 3.3 ABLATION STUDIES

**Model capacity.** We verify our hypothesis that the capacity of the MTL model should not exceed the total capacities of the STL model. We show this on an LSTM model with sentiment analysis tasks. Recall that the capacity of an LSTM model is its output dimension (before the last classification layer). We train an MTL model with all tasks and vary the shared module's capacity to find the optimum from 5 to 500. Similarly we train an STL model for each task and find the optimum.

In Figure 1, we find that the performance of MTL peaks when the shared module has capacity 100. This is much smaller than the total capacities of all the STL models. The result confirms that

---

[4]https://github.com/google-research/bert

Table 1: Comparing the model capacity between MTL and STL.

| Task | STL | | MTL | |
|------|-----|-----|-----|-----|
| | Cap. | Acc. | Cap. | Acc. |
| SST | 200 | 82.3 | | **90.8** |
| MR | 200 | 76.4 | | **96.0** |
| CR | 5 | 73.2 | 100 | **78.7** |
| SUBJ | 200 | 91.5 | | 89.5 |
| MPQA | 500 | 86.7 | | **87.0** |
| TREC | 100 | **85.7** | | 78.7 |
| **Overall** | 1205 | 82.6 | 100 | 85.1 |

Figure 6: Covariance similarity score vs. performance improvements from alignment.

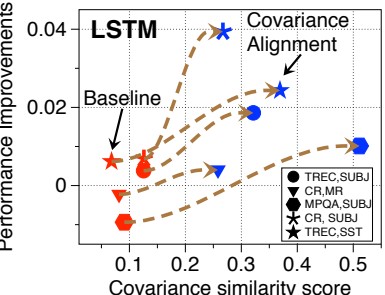

constraining the shared module's capacity is crucial to achieve the ideal performance. Extended results on CNN/MLP to support our hypothesis are shown in Appendix B.5.

**Task covariance.** We apply our metric of task covariance similarity score from Section 2.3 to provide an in-depth study of the covariance alignment method. The hypothesis is that: (a) aligning the covariances helps, which we have shown in Figure 5a; (b) the similarity score between two tasks increases after applying the alignment. We verify the hypothesis on the sentiment analysis tasks. We use the single-task model's embedding before the LSTM layer to compute the covariance.

First, we measure the similarity score using equation 6 between all six single-task models. Then, for each task pair, we train an MTL model using Algorithm 1. We measure the similarity score on the trained MTL model. Our results confirm the hypothesis (Figure 6): (a) we observe increased accuracy on 13 of 15 task pairs by up to 4.1%; (b) the similarity score increases for all 15 task pairs.

**Optimization scheme.** We verify the robustness of Algorithm 2. After selecting two tasks from the ChestX-ray14 dataset, we test our method by assigning random labels to 20% of the data on one task. The labels for the other task remain unchanged.

On 10 randomly selected pairs, our method improves over the unweighted scheme by an average 1.0% AUC score and the techniques of Kendall et al. (2018) by an average 0.4% AUC score. We include more details of this experiment in Appendix B.5.

## 4 RELATED WORK

There has been a large body of recent work on using the multi-task learning approach to train deep neural networks. Liu et al. (2019a); McCann et al. (2018) and subsequent follow-up work show state-of-the-art results on the GLUE benchmark, which inspired our study of an abstraction of the MTL model. Recent work of Zamir et al. (2018); Standley et al. (2019) answer which visual tasks to train together via a heuristic which involves intensive computation. We discuss several lines of studies related to this work. For complete references, we refer the interested readers to the survey of Ruder (2017); Zhang and Yang (2017) and the surveys on domain adaptation and transfer learning by Pan and Yang (2009); Kouw (2018) for references.

**Theoretical studies of multi-task learning.** Of particular relevance to this work are those that study the theory of multi-task learning. The earlier works of Baxter (2000); Ben-David and Schuller (2003) are among the first to formally study the importance of task relatedness for learning multiple tasks. See also the follow-up work of Maurer (2006) which studies generalization bounds of MTL.

A closely related line of work to structural learning is subspace selection, i.e. how to select a common subspace for multiple tasks. Examples from this line work include Obozinski et al. (2010); Wang et al. (2015); Fernando et al. (2013); Elhamifar et al. (2015). Evgeniou and Pontil (2004); Micchelli and Pontil (2005) study a formulation that extends support vector machine to the multi-task setting. See also Argyriou et al. (2008); Pentina et al. (2015); Pentina and Ben-David (2015); Pentina and Lampert (2017) that provide more refined optimization methods and further study. The work of Ben-David et al. (2010) provides theories to measure the differences between source and target tasks for transfer learning in a different model setup. Khodak et al. (2019); Kong et al. (2020);

Du et al. (2020) consider the related meta learning setting, which is in spirit an online setting of multi-task learning.

Our result on restricting the model capacities for multi-task learning is in contrast with recent theoretical studies on over-parametrized models (e.g. Li et al. (2018); Zhang et al. (2019a); Bartlett et al. (2020)), where the model capacities are usually much larger than the regime we consider here. It would be interesting to better understand multi-task learning in the context of over-parametrized models with respect to other phenomenon such as double descent that has been observed in other contexts (Belkin et al. (2019)).

Finally, Zhang et al. (2019b); Shui et al. (2019) consider multi-task learning from the perspective of adversarial robustness. Mahmud and Ray (2008) consider using Kolmogorov complexity measure the effectiveness of transfer learning for decision tree methods.

**Hard parameter sharing vs soft parameter sharing.** The architecture that we study in this work is also known as the hard parameter sharing architecture. There is another kind of architecture called soft parameter sharing. The idea is that each task has its own parameters and modules. The relationships between these parameters are regularized in order to encourage the parameters to be similar. Other architectures that have been studied before include the work of Misra et al. (2016), where the authors explore trainable architectures for convolutional neural networks.

**Domain adaptation.** Another closely related line of work is on domain adaptation. The acute reader may notice the similarity between our study in Section 2.3 and domain adaptation. The crucial difference here is that we are minimizing the multi-task learning objective, whereas in domain adaptation the objective is typically to minimize the objective on the target task. See Ben-David et al. (2010); Zhang et al. (2019b) and the references therein for other related work.

**Optimization techniques.** Guo et al. (2019) use ideas from the multi-armed bandit literature to develop a method for weighting each task. Compared to their method, our SVD-based method is conceptually simpler and requires much less computation. Kendall et al. (2018) derive a weighted loss schme by maximizing a Gaussian likelihood function. Roughly speaking, each task is reweighted by $1/\sigma^2$ where $\sigma$ is the standard deviation of the Gaussian and a penalty of $\log \sigma$ is added to the loss. The values of $\{\sigma_i\}_i$ are also optimized during training. The exact details can be found in the paper. The very recent work of Li and Vasconcelos (2019) show empirical results using a similar idea of covariance normalization on imaging tasks for cross-domain transfer.

## 5 CONCLUSIONS AND FUTURE WORK

We studied the theory of multi-task learning in linear and ReLU-activated settings. We verified our theory and its practical implications through extensive synthetic and real world experiments.

Our work opens up many interesting future questions. First, could we extend the guarantees for choosing optimization schemes to non-linear settings? Second, a limitation of our SVD-based optimization scheduler is that it only applies to settings with the same data. Could we extend the method for heterogeneous task data? More broadly, we hope our work inspires further studies to better understand multi-task learning in neural networks and to guide its practice.

**Acknowledgements.** Thanks to Sharon Y. Li and Avner May for stimulating discussions during early stages of this work. We are grateful to the Stanford StatsML group and the anonymous referees for providing helpful comments that improve the quality of this work. We gratefully acknowledge the support of DARPA under Nos. FA87501720095 (D3M), FA86501827865 (SDH), and FA86501827882 (ASED); NIH under No. U54EB020405 (Mobilize), NSF under Nos. CCF1763315 (Beyond Sparsity), CCF1563078 (Volume to Velocity), and 1937301 (RTML); ONR under No. N000141712266 (Unifying Weak Supervision); the Moore Foundation, NXP, Xilinx, LETI-CEA, Intel, IBM, Microsoft, NEC, Toshiba, TSMC, ARM, Hitachi, BASF, Accenture, Ericsson, Qualcomm, Analog Devices, the Okawa Foundation, American Family Insurance, Google Cloud, Swiss Re, and members of the Stanford DAWN project: Teradata, Facebook, Google, Ant Financial, NEC, VMWare, and Infosys. H. Zhang is supported in part by Gregory Valiant's ONR YIP award (#1704417). The experiments are partly run on Stanford's SOAL cluster. [5] The U.S. Government is authorized to reproduce and distribute reprints for Governmental purposes notwithstanding

---

[5] https://5harad.com/soal-cluster/

any copyright notation thereon. Any opinions, findings, and conclusions or recommendations expressed in this material are those of the authors and do not necessarily reflect the views, policies, or endorsements, either expressed or implied, of DARPA, NIH, ONR, or the U.S. Government.

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

## A    MISSING DETAILS OF SECTION 2

We fill in the missing details left from Section 2. In Section A.1, we provide rigorous arguments regarding the capacity of the shared module. In Section A.2, we fill in the details left from Section 2.3, including the proof of Theorem 2 and its extension to the ReLU model. In Section A.3, we provide the proof of Proposition 3 on the task reweighting schemes. We first describe the notations.

**Notations.** We define the notations to be used later on. We denote $f(x) \lesssim g(x)$ if there exists an absolute constant $C$ such that $f(x) \leq Cg(x)$. The big-O notation $f(x) = O(g(x))$ means that $f(x) \lesssim g(x)$.

Suppose $A \in \mathbb{R}^{m \times n}$, then $\lambda_{\max}(A)$ denotes its largest singular value and $\lambda_{\min}(A)$ denotes its $\min\{m, n\}$-th largest singular value. Alternatively, we have $\lambda_{\min}(A) = \min_{x:\|x\|=1} \|Ax\|$. Let $\kappa(A) = \lambda_{\max}(A)/\lambda_{\min}(A)$ denote the condition number of $A$. Let Id denotes the identity matrix. Let $U^\dagger$ denote the Moore-Penrose pseudo-inverse of the matrix $U$. Let $\|\cdot\|$ denote the Euclidean norm for vectors and spectral norm for matrices. Let $\|\cdot\|_F$ denote the Frobenius norm of a matrix. Let $\langle A, B, =\rangle \operatorname{Tr}(A^\top B)$ denote the inner product of two matrices.

The sine function is define as $\sin(u, v) = \sqrt{1 - \cos(u, v)^2}$, where we assume that $\sin(u, v) \geq 0$ which is without loss of generality for our study.

### A.1    MISSING DETAILS OF SECTION 2.2

We describe the full detail to show that our model setup captures the phenomenon that the shared module should be smaller than the sum of capacities of the single-task models. We state the following proposition which shows that the quality of the subspace $B$ in equation 1 determines the performance of multi-task learning. This supplements the result of Proposition 1.

**Proposition 4.** *In the optimum of $f(\cdot)$ (equation 1), each $A_i$ selects the vector $v$ within the column span of $g_B(X_i)$ to minimize $L(v, y_i)$. As a corollary, in the linear setting, the optimal $B$ can be achieved at a rotation matrix $B^\star \subseteq \mathbb{R}^{d \times r}$ by maximizing*

$$\sum_{i=1}^{k} \langle B(B^\top X_i^\top X_i B)^\dagger B^\top, X_i^\top y_i y_i^\top X_i \rangle. \tag{7}$$

*Furthermore, any $B^\star$ which contains $\{\theta_i\}_{i=1}^{k}$ in its column subspace is optimal. In particular, for such a $B^\star$, there exists $\{A_i^\star\}$ so that $B^\star A_i^\star = \theta_i$ for all $1 \leq i \leq k$.*

*Proof.* Recall the MTL objective in the linear setting from equation 3 as follows:

$$\min f(A_1, A_2, \ldots, A_k; B) = \sum_{i=1}^{k} (X_i B A_i - y_i)^2,$$

Note that the linear layer $A_i$ can pick any combination within the subspace of $B$. Therefore, we could assume without loss of generality that $B$ is a rotation matrix. i.e. $B^\top B = \operatorname{Id}$. After fixing $B$, since objective $f(\cdot)$ is linear in $A_i$ for all $i$, by the local optimality condition, we obtain that

$$A_i = (B^\top X_i^\top X_i B)^\dagger B^\top X_i^\top y_i$$

Replacing the solution of $A_i$ to $f(\cdot)$, we obtain an objective over $B$.

$$h(B) = \sum_{i=1}^{k} \|X_i B(B^\top X_i^\top X_i B)^\dagger B^\top X_i^\top y_i - y_i\|_F^2.$$

Next, note that

$$\|X_i B(B^\top X_i^\top X_i B)^\dagger B^\top X_i^\top y_i\|_F^2 = \operatorname{Tr}(y_i^\top X_i B(B^\top X_i^\top X_i B)^\dagger B^\top X_i^\top y_i)$$
$$= \langle B(B^\top X_i^\top X_i B)B^\top, X_i^\top y_i y_i^\top X_i \rangle,$$

where we used the fact that $A^\dagger A A^\dagger = A^\dagger$ for $A = B^\top X_i^\top X_i B$ in the first equation. Hence we have shown equation 7.

For the final claim, as long as $B^\star$ contains $\{\theta_i\}_{i=1}^k$ in its column subspace, then there exists $A_i^\star$ such that $B^\star A_i^\star = \theta_i$. The $B^\star$ and $\{A_i^\star\}_{i=1}^k$ are optimal solutions because each $\theta_i$ is an optimal solution for the single-task problem. $\qquad\square$

The above result on linear regression suggests the intuition that optimizing an MTL model reduces to optimizing over the span of $B$. The intuition can be easily extended to linear classification tasks as well as mixtures of regression and classification tasks.

**Extension to the ReLU setting.** If the shared module's capacity is larger than the total capacities of the STL models, then we can put all the STL model parameters into the shared module. As in the linear setting, the final output layer $A_i$ can pick out the optimal parameter for the $i$-th task. This remains an optimal solution to the MTL problem in the ReLU setting. Furthermore, there is no transfer between any two tasks through the shared module.

## A.2 Missing Details of Section 2.3

### A.2.1 The Effect of Cosine Similarity

We consider the effect of varying the cosine similarity between single task models in multi-task learning. We first describe the following proposition to solve the multi-task learning objective when the covariances of the task data are the same. The idea is similar to the work of Ando and Zhang (2005) and we adapt it here for our study.

**Proposition 5.** *Consider the reweighted loss of equation 2 with the encoding function being linear, where the weights are $\{\alpha_i\}_{i=1}^k$. Suppose the task features of every task have the same covariance: $X_i^\top X_i = \Sigma$ for all $1 \le i \le k$. Let $\Sigma = VDV^\top$ be the singular vector decomposition (SVD) of $\Sigma$. Then the optimum of $f(\cdot)$ in equation 3 is achieved at:*

$$B^\star = VD^{-1/2}C^\star,$$

*where $C^\star C^{\star\top}$ is the best rank-$r$ approximation subspace of $\sum_{i=1}^k \alpha_i U_i^\top y_i y_i^\top U_i$ and $X_i = U_i DV^\top$ is the SVD of $X_i$, for each $1 \le i \le k$.*

*As a corollary, denote by $\lambda_1, \lambda_2, \ldots, \lambda_k$ as the singular values of $D^{-1}V^\top \sum_{i=1}^k \alpha_i X_i^\top y_i y_i^\top X_i$ in decreasing order. Then the difference between an MTL model with hidden dimension $r$ and the all the single task models is bounded by $\sum_{i=r+1}^k \lambda_i^2$.*

*Proof.* Note that $B^\star$ is obtained by maximizing

$$\sum_{i=1}^k \langle B(B^\top X_i^\top X_i B)^{-1}B^\top, \alpha_i X_i^\top y_i y_i^\top X_i \rangle$$

Let $C = DV^\top B$. Clearly, there is a one to one mapping between $B$ and $C$. And we have $B = VD^{-1}C$. Hence the above is equivalent to maximizing over $C \subseteq \mathbb{R}^{d\times r}$ with

$$\sum_{i=1}^k \langle C(C^\top C)^{-1}C^\top, D^{-1}V^\top \left(\sum_{i=1}^k \alpha_i X_i^\top y_i y_i^\top X_i\right) VD^{-1} \rangle$$

$$= \langle C(C^\top C)^{-1}C^\top, \sum_{i=1}^k \alpha_i U_i^\top y_i y_i^\top U_i \rangle.$$

Note that $C(C^\top C)^{-1}C^\top$ is a projection matrix onto a subspace of dimension $r$. Hence the maximum (denote by $C^\star$) is attained at the best rank-$r$ approximation subspace of $\sum_{i=1}^k \alpha_i U_i^\top y_i y_i^\top U_i$. $\qquad\square$

To illustrate the above proposition, consider a simple setting where $X_i$ is identity for every $1 \le i \le k$, and $y_i = e_i$, i.e. the $i$-th basis vector. Note that the optimal solution for the $i$-th task is $(X_i^\top X_i)^{-1}X_i^\top y_i = y_i$. Hence the optimal solutions are orthogonal to each other for all the tasks, with $\lambda_i = 1$ for all $1 \le i \le k$. And the minimum STL error is zero for all tasks.

Consider the MTL model with hidden dimension $r$. By Proposition 5, the minimum MTL error is achieved by the best rank-$r$ approximation subspace to $\sum_{i=1}^k X_i^\top y_i y_i^\top X_i = \sum_{i=1}^k y_i y_i^\top$. Denote the optimum as $B_r^\star$. The MTL error is:

$$\sum_{i=1}^k \|y_i\|^2 - \langle \sum_{i=1}^k y_i y_i^\top, B_r^\star B_r^{\star\top} \rangle = k - r.$$

**Different data covariance.** We provide upper bounds on the quality of MTL solutions for different data covariance, which depend on the relatedness of all the tasks. The following procedure gives the precise statement. Consider $k$ regression tasks with data $\{(X_i, y_i)\}_{i=1}^k$. Let $\theta_i = (X_i^\top X_i)^\dagger X_i^\top y_i$ denote the optimal solution of each regression task. Let $W \subseteq \mathbb{R}^{d \times k}$ denote the matrix where the $i$-th column is equal to $\theta_i$. Consider the following procedure for orthogonalizing $W$ for $1 \le i \le k$.

a) Let $W_i^\star \in \mathbb{R}^d$ denote the vector which maximizes $\sum_{i=1}^k \langle \frac{X_i B}{\|X_i B\|}, y_i \rangle^2$ over $B \in \mathbb{R}^d$;

b) Denote by $\lambda_j = \sum_{j=1}^k \langle \frac{X_j W_j^\star}{\|X_j W_j^\star\|}, y_j \rangle^2$;

c) For each $1 \le i \le k$, project $X_i W_i^\star$ off from every column of $X_i$. Go to Step a).

**Proposition 6.** *Suppose that $r \le d$. Let $B^\star$ denote the optimal MTL solution of capacity $r$ in the shared module. Denote by $OPT = \sum_{i=1}^k (\|y_i\|^2 - \|X_i(X_i^\top X_i)^\dagger X_i^\top y_i\|^2)$. Then $h(B^\star) \le OPT - \sum_{i=r+1}^d \lambda_i$.*

*Proof.* It suffices to show that $OPT$ is equal to $\sum_{i=1}^k \lambda_i$. The result then follows since $h(B^\star)$ is less than the error given by $W_1^\star, \ldots, W_k^\star$, which is equal to $OPT - \sum_{i=r+1}^d \lambda_i$. □

### A.2.2 Proof of Theorem 2

We fill in the proof of Theorem 2. First, we restate the result rigorously as follows.

**Theorem 2.** *For $i = 1, 2$, let $(X_i, y_i) \in (\mathbb{R}^{m_i \times d}, \mathbb{R}^{m_i})$ denote two linear regression tasks with parameters $\theta_i \in \mathbb{R}^d$. Suppose that each row of $X_1$ is drawn independently from a distribution with covariance $\Sigma_1 \subseteq \mathbb{R}^{d \times d}$ and bounded $l_2$-norm $\sqrt{L}$. Assume that $\theta_1^\top \Sigma_1 \theta_1 = 1$ w.l.o.g.*

*Let $c \in [\kappa(X_2)\sin(\theta_1, \theta_2), 1/3]$ denote the desired error margin. Denote by $(B^\star, A_1^\star, A_2^\star)$ the optimal MTL solution. With probability $1 - \delta$ over the randomness of $(X_1, y_1)$, when*

$$m_1 \gtrsim \max\left( \frac{L\|\Sigma_1\|\log\frac{d}{\delta}}{\lambda_{\min}^2(\Sigma_1)}, \frac{\kappa(\Sigma_1)\kappa^2(X_2)}{c^2}\|y_2\|^2, \frac{\kappa^2(\Sigma_1)\kappa^4(X_2)}{c^4}\sigma_1^2 \log\frac{1}{\delta} \right),$$

*we have that $\|B^\star A_2^\star - \theta_2\|/\|\theta_2\| \le 6c + \frac{1}{1-3c}\|\varepsilon_2\|/\|X_2\theta_2\|$.*

We make several remarks to provide more insight on Theorem 2.

- Theorem 2 guarantees positive transfers in MTL, when the source and target models are close and the number of source samples is large. While the intuition is folklore in MTL, we provide a formal justification in the linear and ReLU models to quantify the phenomenon.

- The error bound decreases with $c$, hence the smaller $c$ is the better. On the other hand, the required number of data points $m_1$ increases. Hence there is a trade-off between accuracy and the amount of data.

- $c$ is assumed to be at most $1/3$. This assumption arises when we deal with the label noise of task 2. If there is no noise for task 2, then this assumption is not needed. If there is noise for task 2, this assumption is satisfied when $\sin(\theta_1, \theta_2)$ is less than $1/(3\kappa(X_2))$. In synthetic experiments, we observe that the dependence on $\kappa(X_2)$ and $\sin(\theta_1, \theta_2)$ both arise in the performance of task 2, cf. Figure 3 and Figure 7, respectively.

The proof of Theorem 2 consists of two steps.

a) We show that the angle between $B^\star$ and $\theta_1$ will be small. Once this is established, we get a bound on the angle between $B^\star$ and $\theta_2$ via the triangle inequality.

b) We bound the distance between $B^\star A_2$ and $\theta_2$. The distance consists of two parts. One part comes from $B^\star$, i.e. the angle between $B^\star$ and $\theta_2$. The second part comes from $A_2$, i.e. the estimation error of the norm of $\theta_2$, which involves the signal to noise ratio of task two.

We first show the following geometric fact, which will be used later in the proof.

**Fact 7.** *Let $a, b \in \mathbb{R}^d$ denote two unit vectors. Suppose that $X \in \mathbb{R}^{m \times d}$ has full column rank with condition number denoted by $\kappa = \kappa(X)$. Then we have*

$$|\sin(Xa, Xb)| \geq \frac{1}{\kappa^2} |\sin(a, b)|.$$

*Proof.* Let $X = UDV^\top$ be the SVD of $X$. Since $X$ has full column rank by assumption, we have $X^\top X = XX^\top = \mathrm{Id}$. Clearly, we have $\sin(Xa, Xb) = \sin(DV^\top a, DV^\top b)$. Denote by $a' = V^\top a$ and $b' = V^\top b$. We also have that $a'$ and $b'$ are both unit vectors, and $\sin(a', b') = \sin(a, b)$. Let $\lambda_1, \ldots, \lambda_d$ denote the singular values of $X$. Then,

$$\begin{aligned}
\sin^2(Da', Db') &= 1 - \frac{\left(\sum_{i=1}^d \lambda_i^2 a_i' b_i'\right)^2}{\left(\sum_{i=1}^d \lambda_i^2 a_i'^2\right)\left(\sum_{i=1}^d \lambda_i^2 b_i'^2\right)} \\
&= \frac{\sum_{1 \leq i,j \leq d} \lambda_i^2 \lambda_j^2 (a_i' b_j' - a_j' b_i')^2}{\left(\sum_{i=1}^d \lambda_i^2 a_i'^2\right)\left(\sum_{i=1}^d \lambda_j^2 b_i'^2\right)} \\
&\geq \frac{\lambda_{\min}^4}{\lambda_{\max}^4} \cdot \sum_{1 \leq i,j \leq d} (a_i' b_j' - a_j' b_i')^2 \\
&= \frac{1}{\kappa^4}((\sum_{i=1}^d a_i'^2)(\sum_{i=1}^d b_i'^2) - (\sum_{i=1}^d a_i' b_i')^2) = \frac{1}{\kappa^4}\sin^2(a', b').
\end{aligned}$$

This concludes the proof. $\qquad\qquad\qquad\qquad\qquad\qquad\qquad\qquad\qquad\qquad\qquad\square$

We first show the following Lemma, which bounds the angle between $B^\star$ and $\theta_2$.

**Lemma 8.** *In the setting of Theorem 2, with probability $1 - \delta$ over the randomness of task one, we have that*

$$|\sin(B^\star, \theta_2)| \leq \sin(\theta_1, \theta_2) + c/\kappa(X_2).$$

*Proof.* We note that $h(B^\star) \geq \|y_1\|^2$ by the optimality of $B^\star$. Furthermore, $\langle \frac{X_2 B^\star}{\|X_2 B^\star\|}, y_2 \rangle \leq \|y_2\|^2$. Hence we obtain that

$$\langle \frac{X_1 B^\star}{\|X_1 B^\star\|}, y_1 \rangle^2 \geq \|y_1\|^2 - \|y_2\|^2.$$

For the left hand side,

$$\begin{aligned}
\langle \frac{X_1 B^\star}{\|X_1 B^\star\|}, y_1 \rangle^2 &= \langle \frac{X_1 B^\star}{\|X_1 B^\star\|}, X_1 \theta_1 + \varepsilon_1 \rangle^2 \\
&= \langle \frac{X_1 B^\star}{\|X_1 B^\star\|}, X_1 \theta_1 \rangle^2 + \langle \frac{X_1 B^\star}{\|X_1 B^\star\|}, \varepsilon_1 \rangle^2 + 2\langle \frac{X_1 B^\star}{\|X_1 B^\star\|}, X_1 \theta_1 \rangle \langle \frac{X_1 B^\star}{\|X_1 B^\star\|}, \varepsilon_1 \rangle
\end{aligned}$$

Note that the second term is a chi-squared random variable with expectation $\sigma_1^2$. Hence it is bounded by $\sigma_1^2 \sqrt{\log \frac{1}{\delta}}$ with probability at least $1 - \delta$. Similarly, the third term is bounded by $2\|X_1 \theta_1\| \sigma_1 \sqrt{\log \frac{1}{\delta}}$ with probability $1 - \delta$. Therefore, we obtain the following:

$$\|X_1 \theta_1\|^2 \cos^2(X_1 B^\star, X_1 \theta_1) \geq \|y_1\|^2 - \|y_2\|^2 - (\sigma_1^2 + 2\sigma_1 \|X_1 \theta_1\|)\sqrt{\log \frac{1}{\delta}}$$

Note that

$$\|y_1\|^2 \geq \|X_1\theta_1\|^2 + 2\langle X_1\theta_1, \varepsilon_1\rangle$$
$$\geq \|X_1\theta_1\|^2 - 2\|X_1\theta_1\|\sigma_1\sqrt{\log\frac{1}{\delta}}.$$

Therefore,

$$\|X_1\theta_1\|^2 \cos^2(X_1B^\star, X_1\theta_1) \geq \|X_1\theta_1\|^2 - \|y_2\|^2 - (\sigma_1^2 + 3\sigma_1\|X_1\theta_1\|)\sqrt{log\frac{1}{\delta}}$$

$$\Rightarrow \sin^2(X_1B^\star, X_1\theta_1) \leq \frac{\|y_2\|^2}{\|X_1\theta_1\|^2} + \frac{4\sigma_1\sqrt{\log\frac{1}{\delta}}}{\|X_1\theta_1\|}$$

$$\Rightarrow \sin^2(B^\star, \theta_1) \leq \kappa^2(X_1)\left(\frac{\|y_2\|^2}{\|X_1\theta_1\|^2} + \frac{4\sigma_1\sqrt{\log\frac{1}{\delta}}}{\|X_1\theta_1\|}\right) \qquad \text{(by Lemma 7)}$$

By matrix Bernstein inequality (see e.g. Tropp et al. (2015)), when $m_1 \geq 10\|\Sigma_1\|\log\frac{d}{\delta}/\lambda_{\min}^2(\Sigma_1)$, we have that:

$$\left\|\frac{1}{m_1}X_1^\top X_1 - \Sigma_1\right\| \leq \frac{1}{2}\lambda_{\min}(\Sigma_1).$$

Hence we obtain that $\kappa^2(X_1) \leq 3\kappa(\Sigma_1)$ and $\|X_1\theta_1\|^2 \geq m_1 \cdot \theta_1^\top\Sigma_1\theta_1/2 \geq m_1/2$ (where we assumed that $\theta_1^\top\Sigma_1\theta_1 = 1$). Therefore,

$$\sin^2(B^\star, \theta_1) \leq 3\kappa(\Sigma_1)\left(\frac{\|y_2\|^2}{m_1^2/4} + \frac{4\sigma_1\sqrt{\log\frac{1}{\delta}}}{\sqrt{m_1/2}}\right),$$

which is at most $c^2/\kappa^2(X_2)$ by our setting of $m_1$. Therefore, the conclusion follows by triangle inequality (noting that both $c$ and $\sin(\theta_1, \theta_2)$ are less than $1/2$). $\qquad\square$

Based on the above Lemma, we are now to ready to prove Theorem 2.

*Proof of Theorem 2.* Note that in the MTL model, after obtaining $B^\star$, we then solve the linear layer for each task. For task 2, this gives weight value $A_2^\star := \langle X_2\hat{\theta}, y_2\rangle/\|X_2\hat{\theta}\|^2$. Thus the regression coefficients for task 2 is $B^\star A_2^\star$. For the rest of the proof, we focus on bounding the distance between $B^\star A_2^\star$ and $\theta_2$. By triangle inequality,

$$\|B^\star A_2^\star - \theta_2\| \leq \frac{|\langle X_2B^\star, \varepsilon_2\rangle|}{\|X_2B^\star\|^2} + \left|\frac{\langle X_2B^\star, X_2\theta_2\rangle}{\|X_2B^\star\|^2} - \|\theta_2\|\right| + \|B^\star\|\theta_2\| - \theta_2\|. \qquad (8)$$

Note that the second term of equation 8 is equal to

$$\frac{|\langle X_2B^\star, X_2(\theta_2 - \|\theta_2\|B^\star)\rangle|}{\|X_2B^\star\|^2} \leq \kappa(X_2) \cdot \|\theta_2 - \|\theta_2\|B^\star\|.$$

The first term of equation 8 is bounded by

$$\frac{\|\varepsilon_2\|}{\|X_2B^\star\|} \leq \frac{\|\varepsilon_2\|\|\theta_2\|}{\|X_2\theta_2\| - \|X_2(\theta_2 - \|\theta_2\|B^\star)\|}. \qquad (9)$$

Lastly, we have that

$$\|\theta_2 - \|\theta_2\|B^\star\|^2 = \|\theta_2\|^2 2(1 - \cos(B^\star, \theta_2)) \leq 2\|\theta_2\|^2 \sin^2(B^\star, \theta_2)$$

By Lemma 8, we have

$$|\sin(B^\star, \theta_2)| \leq \sin(\theta_1, \theta_2) + c/\kappa(X_2)$$

Therefore, we conclude that equation 9 is at most

$$\frac{\|\varepsilon_2\| \cdot \|\theta_2\|}{\|X_2\theta_2\| - \sqrt{2}\lambda_{\max}(X_2)\|\theta_2\|\sin(\theta_1, \theta_2) - \sqrt{2}c\lambda_{\min}(X_2)\|\theta_2\|}$$

$$\leq \frac{\|\varepsilon_2\| \cdot \|\theta_2\|}{\|X_2\theta_2\| - 3c\lambda_{\min}(X_2)\|\theta_2\|}$$

$$\leq \frac{1}{1-3c}\frac{\|\varepsilon_2\| \cdot \|\theta_2\|}{\|X_2\theta_2\|}$$

Thus equation 8 is at most the following.

$$\|\theta_2\| \cdot \left(\frac{1}{1-3c}\frac{\|\varepsilon_2\|}{\|X_2\theta_2\|} + \sqrt{2}(\kappa(X_2)+1) \cdot \sin(B^\star, \theta_2)\right)$$

$$\leq \|\theta_2\| \cdot \left(\frac{1}{1-3c}\frac{\|\varepsilon_2\|}{\|X_2\theta_2\|} + 6c\right).$$

Hence we obtain the desired estimation error of $BA_2^\star$. $\qquad\square$

### A.2.3 EXTENSION TO THE RELU MODEL

In this part, we extend Theorem 2 to the ReLU model. Note that the problem is reduced to the following objective.

$$\max_{B \in \mathbb{R}^d} g(B) = \langle \frac{\text{ReLU}(X_1 B)}{\|\text{ReLU}(X_1 B)\|}, y_1 \rangle^2 + \langle \frac{\text{ReLU}(X_2 B)}{\|\text{ReLU}(X_2 B)\|}, y_2 \rangle^2 \tag{10}$$

We make a crucial assumption that task 1's input $X_1$ follows the Gaussian distribution. Note that making distributional assumptions is necessary because for worst-case inputs, even optimizing a single ReLU function under the squared loss is NP-hard (Manurangsi and Reichman (2018)). We state our result formally as follows.

**Theorem 9.** *Let* $(X_1, y_1) \in (\mathbb{R}^{m_1 \times d}, \mathbb{R}^{m_1})$ *and* $(X_2, y_2) \in (\mathbb{R}^{m_2 \times d}, \mathbb{R}^{m_2})$ *denote two tasks. Suppose that each row of* $X_1$ *is drawn from the standard Gaussian distribution. And* $y_i = a_i \cdot \text{ReLU}(X_i\theta_i) + \varepsilon_i$ *are generated via the ReLU model with* $\theta_1, \theta_2 \in \mathbb{R}^d$. *Let* $\mathbb{E}\left[(a_i \cdot \text{ReLU}(X_i\theta_i))_j^2\right] = 1$ *for every* $1 \leq j \leq m_1$ *without loss of generality, and let* $\sigma_1^2$ *denote the variance of every entry of* $\varepsilon_1$.

*Suppose that* $c \geq \sin(\theta_1, \theta_2)/\kappa(X_2)$. *Denote by* $(B^\star, A_1^\star, A_2^\star)$ *the optimal MTL solution of equation 10. With probability* $1 - \delta$ *over the randomness of* $(X_1, y_1)$, *when*

$$m_1 \gtrsim \max\left(\frac{d\log d}{c^2}(\frac{1}{c^2} + \log d), \frac{\|y_2\|^2}{c^2}\right),$$

*we have that the estimation error is at most:*

$$\sin(B^\star, \theta_1) \leq \sin(\theta_1, \theta_2) + O(c/\kappa(X_2)),$$

$$\frac{|A_2^\star - a_2|}{a_2} \leq O(c) + \frac{1}{(1-O(c))} \cdot \frac{\|\varepsilon_2\|}{a_2 \cdot \text{ReLU}(\|X_2\theta_2\|)}$$

*Proof.* The proof follows a similar structure to that of Theorem 2. Without loss of generality, we can assume that $\theta_1, \theta_2$ are both unit vectors. We first bound the angle between $B^\star$ and $\theta_1$.

By the optimality of $B^\star$, we have that:

$$\langle \frac{\text{ReLU}(X_1 B^\star)}{\|\text{ReLU}(X_1 B^\star)\|}, y_1 \rangle^2 \geq \langle \frac{\text{ReLU}(X_1 \theta_1)}{\|\text{ReLU}(X_1 \theta_1)\|}, y_1 \rangle^2 - \|y_2\|^2$$

From this we obtain:

$$a_1^2 \cdot \langle \frac{\text{ReLU}(X_1 B^\star)}{\|\text{ReLU}(X_1 B^\star)\|}, \text{ReLU}(X_1 B^\star) \rangle^2$$

$$\geq a_1^2 \cdot \|\text{ReLU}(X_1 \theta_1)\|^2 - \|y_2\|^2 - (\sigma_1^2 + 4a_1 \cdot \sigma_1 \|\text{ReLU}(X_1 \theta_1)\|)\sqrt{\log\frac{1}{\delta}} \tag{11}$$

Note that each entry of $\text{ReLU}(X_1\theta_1)$ is a truncated Gaussian random variable. By the Hoeffding bound, with probability $1 - \delta$ we have

$$\left| \|\text{ReLU}(X_1\theta_1)\|^2 - \frac{m_1}{2} \right| \leq \sqrt{\frac{m_1}{2} \log \frac{1}{\delta}}.$$

As for $\langle \text{ReLU}(X_1 B^\star), \text{ReLU}(X_1\theta_1) \rangle$, we will use an epsilon-net argument over $B^\star$ to show the concentration. For a fixed $B^\star$, we note that this is a sum of independent random variables that are all bounded within $O(\log \frac{m_1}{\delta})$ with probability $1 - \delta$. Denote by $\phi$ the angle between $B^\star$ and $\theta_1$, a standard geometric fact states that (see e.g. Lemma 1 of Du et al. (2017)) for a random Gaussian vector $x \in \mathbb{R}^d$,

$$\mathbb{E}_x \left[ \text{ReLU}(x^\top B^\star) \cdot \text{ReLU}(x^\top \theta_1) \right] = \frac{\cos \phi}{2} + \frac{\cos \phi (\tan \phi - \phi)}{2\pi} := \frac{g(\phi)}{2}.$$

Therefore, by applying Bernstein's inequality and union bound, with probability $1 - \eta$ we have:

$$|\langle \text{ReLU}(X_1 B^\star), \text{ReLU}(X_1\theta_1) \rangle - m_1 g(\phi)/2| \leq 2\sqrt{m_1 g(\phi) \log \frac{1}{\eta}} + \frac{2}{3} \log \frac{1}{\eta} \log \frac{m_1}{\delta}$$

By standard arguments, there exists a set of $d^{O(d)}$ unit vectors $S$ such that for any other unit vector $u$ there exists $\hat{u} \in S$ such that $\|u - \hat{u}\| \leq \min(1/d^3, c^2/\kappa^2(X_2))$. By setting $\eta = d^{-O(d)}$ and take union bound over all unit vectors in $S$, we have that there exists $\hat{u} \in S$ satisfying $\|B^\star - \hat{u}\| \leq \min(1/d^3, c^2/\kappa^2(X_2))$ and the following:

$$|\langle \text{ReLU}(X_1 \hat{u}), \text{ReLU}(X_1\theta_1) \rangle - m_1 g(\phi')/2| \lesssim \sqrt{m_1 d \log d} + d \log^2 d$$
$$\leq 2m_1 c^2/\kappa^2(X_2) \qquad \text{(by our setting of } m_1\text{)}$$

where $\phi'$ is the angle between $\hat{u}$ and $\theta_1$. Note that

$$\left| \langle \text{ReLU}(X_1 \hat{\theta}) - \text{ReLU}(X_1 B^\star), \text{ReLU}(X_1\theta_1) \rangle \right| \leq \|X_1(\hat{u} - B^\star)\| \cdot \|\text{ReLU}(X_1\theta_1)\|$$
$$\leq c^2/\kappa^2(X_2) \cdot O(m_1)$$

Together we have shown that

$$|\langle \text{ReLU}(X_1 B^\star), \text{ReLU}(X_1\theta_1) \rangle - m_1 g(\phi')/2| \leq c^2/\kappa^2(X_2) \cdot O(m_1).$$

Combined with equation 11, by our setting of $m_1$, it is not hard to show that

$$g(\phi') \geq 1 - O(c^2/\kappa^2(X_2)).$$

Note that

$$1 - g(\phi') = 1 - \cos \phi' - \cos \phi'(\tan \phi' - \phi')$$
$$\leq 1 - \cos \phi' = 2\sin^2 \frac{\phi'}{2} \lesssim c^2/\kappa^2(X_2),$$

which implies that $\sin^2 \phi' \lesssim c^2/\kappa^2(X_2)$ (since $\cos \frac{\phi'}{2} \geq 0.9$). Finally note that $\|\hat{u} - B^\star\| \leq c^2/\kappa^2(X_2)$, hence

$$\|\hat{u} - B^\star\|^2 = 2(1 - \cos(\hat{u}, B^\star)) \geq 2\sin^2(\hat{u}, B^\star).$$

Overall, we conclude that $\sin(B^\star, \theta_1) \leq O(c/\kappa(X_2))$. Hence

$$\sin(B^\star, \theta_2) \leq \sin(\theta_1, \theta_2) + O(c/\kappa(X_2)).$$

For the estimation of $a_2$, we have

$$\left| \frac{\langle \text{ReLU}(X_2 B^\star), y_2 \rangle}{\|\text{ReLU}(X_2 B^\star)\|^2} - a_2 \right| \leq \frac{|\langle \text{ReLU}(X_2 B^\star), \varepsilon_2 \rangle|}{\|\text{ReLU}(X_2 B^\star)\|^2}$$
$$+ a_2 \left| \frac{\langle \text{ReLU}(X_2 B^\star), \text{ReLU}(X_2 B^\star) - \text{ReLU}(X_2\theta_2) \rangle}{\|\text{ReLU}(X_2 B^\star)\|^2} \right|$$

The first part is at most

$$\frac{\|\varepsilon_2\|}{\|\text{ReLU}(X_2 B^\star)\|} \leq \frac{\|\varepsilon_2\|}{\|\text{ReLU}(X_2\theta_2)\| - \|\text{ReLU}(X_2\theta_2) - \text{ReLU}(X_2 B^\star)\|}$$
$$\leq \frac{1}{1 - O(c)} \frac{\|\varepsilon_2\|}{\|\text{ReLU}(X_2\theta_2)\|}$$

Similarly, we can show that the second part is at most $O(c)$. Therefore, the proof is complete. $\qquad \square$

A.3    PROOF OF PROPOSITION 3

In this part, we present the proof of Proposition 3. In fact, we present a more refined result, by showing that all local minima are global minima for the reweighted loss in the linear case.

$$f(A_1, A_2, \ldots, A_k; B) = \sum_{i=1}^{k} \alpha_i \|X_i B A_i - y_i\|_F^2. \tag{12}$$

The key is to reduce the MTL objective $f(\cdot)$ to low rank matrix approximation, and apply recent results by Balcan et al. (2018) which show that there is no spurious local minima for the latter problem .

**Lemma 10.** *Assume that $X_i^\top X_i = \alpha_i \Sigma$ with $\alpha_i > 0$ for all $1 \leq i \leq k$. Then all the local minima of $f(A_1, \ldots, A_k; B)$ are global minima of equation 3.*

*Proof.* We first transform the problem from the space of $B$ to the space of $C$. Note that this is without loss of generality, since there is a one to one mapping between $B$ and $C$ with $C = DV^\top B$. In this case, the corresponding objective becomes the following.

$$
\begin{aligned}
g(A_1, \ldots, A_k; B) &= \sum_{i=1}^{k} \alpha_i \cdot \|U_i C A_i - y_i\|^2 \\
&= \sum_{i=1}^{k} \|C(\sqrt{\alpha_i} A_i) - \sqrt{\alpha_i} U_i^\top y_i\|^2 + \sum_{i=1}^{k} \alpha_i \cdot (\|y_i\|^2 - \|U_i^\top y_i\|^2)
\end{aligned}
$$

The latter expression is a constant. Hence it does not affect the optimization solution. For the former, denote by $A \in \mathbb{R}^{r \times k}$ as stacking the $\sqrt{\alpha_i} A_i$'s together column-wise. Similarly, denote by $Z \in \mathbb{R}^{d \times k}$ as stacking $\sqrt{\alpha_i} U_i^\top y_i$ together column-wise. Then minimizing $g(\cdot)$ reduces solving low rank matrix approximation: $\|CA - Z\|_F^2$.

By Lemma 3.1 of Balcan et al. (2018), the only local minima of $\|CA - Z\|_F^2$ are the ones where $CA$ is equal to the best rank-$r$ approximation of $Z$. Hence the proof is complete.    $\square$

Now we are ready to prove Proposition 3.

*Proof of Proposition 3.* By Proposition 5, the optimal solution of $B^\star$ for equation 12 is $VD^{-1}$ times the best rank-$r$ approximation to $\alpha_i U^\top y_i y_i^\top U$, where we denote the SVD of $X$ as $UDV^\top$. Denote by $Q_r Q_r^\top$ as the best rank-$r$ approximation to $U^\top ZZ^\top U$, where we denote by $Z = [\sqrt{\alpha_1} y_1, \sqrt{\alpha_2} y_2, \ldots, \sqrt{\alpha_k} y_k]$ as stacking the $k$ vectors to a $d$ by $k$ matrix. Hence the result of Proposition 5 shows that the optimal solution $B^\star$ is $VD^{-1}Q_r$, which is equal to $(X^\top X)^{-1}XQ^r$. By Proposition 4, the optimality of $B^\star$ is the same up to transformations on the column space. Hence the proof is complete.    $\square$

To show that all local minima are also equal to $(X^\top X)^{-1}XQ^r$, we can simply apply Lemma 10 and Proposition 3.

**Remark.** This result only applies to the linear model and does not work on ReLU models. The question of characterizing the optimization landscape in non-linear ReLU models is not well-understood based on the current theoretical understanding of neural networks. We leave this for future work.

## B  SUPPLEMENTARY EXPERIMENTAL RESULTS

We fill in the details left from our experimental section. In Appendix B.1, we review the datasets used in our experiments. In Appendix B.2, we describe the models we use on each dataset. In Appendix B.3, we describe the training procedures for all experiments. In Appendix B.4 and Appendix B.5, we show extended synthetic and real world experiments to support our claims.

### B.1  DATASETS

We describe the synthetic settings and the datasets *Sentiment Analysis*, *General Language Understanding Evaluation (GLUE) benchmark*, and *ChestX-ray14* used in the experiments.

**Synthetic settings.** For the synthetic experiments, we draw 10,000 random data samples with dimension $d = 100$ from the standard Gaussian $\mathcal{N}(0, 1)$ and calculate the corresponding labels based on the model described in experiment. We split the data samples into training and validation sets with 9,000 and 1,000 samples in each. For classification tasks, we generate the labels by applying a sigmoid function and then thresholding the value to binary labels at $0.5$. For ReLU regression tasks, we apply the ReLU activation function on the real-valued labels. The number of data samples used in the experiments varies depending on the specification. Specifically, for the task covariance experiment of Figure 3, we fix task 1's data with $m_1 = 9,000$ training data and vary task 2's data under three settings: (i) same rotation $Q_1 = Q_2$ but different singular values $D_1 \neq D_2$; (ii) same singular values $D_1 = D_2$ but random rotations $Q_1 \neq Q_2$.

**Sentiment analysis.** For the sentiment analysis task, the goal is to understand the sentiment opinions expressed in the text based on the context provided. This is a popular text classification task which is usually formulated as a multi-label classification task over different ratings such as positive (+1), negative (-1), or neutral (0). We use six sentiment analysis benchmarks in our experiments:

- Movie review sentiment (MR): In the MR dataset (Pang and Lee (2005)), each movie review consists of a single sentence. The goal is to detect positive vs. negative reviews.

- Sentence subjectivity (SUBJ): The SUBJ dataset is proposed in Pang and Lee (2004) and the goal is to classify whether a given sentence is subjective or objective.

- Customer reviews polarity (CR): The CR dataset (Hu and Liu (2004)) provides customer reviews of various products. The goal is to categorize positive and negative reviews.

- Question type (TREC): The TREC dataset is collected by Li and Roth (2002). The aim is to classify a question into 6 question types.

- Opinion polarity (MPQA): The MPQA dataset detects whether an opinion is polarized or not (Wiebe et al. (2005)).

- Stanford sentiment treebank (SST): The SST dataset, created by Socher et al. (2013), is an extension of the MR dataset.

**The General Language Understanding Evaluation (GLUE) benchmark.** GLUE is a collection of NLP tasks including question answering, sentiment analysis, text similarity and textual entailment problems. The GLUE benchmark is a state-of-the-art MTL benchmark for both academia and industry. We select five representative tasks including CoLA, MRPC, QNLI, RTE, and SST-2 to validate our proposed method. We emphasize that the goal of this work is not to come up with a state-of-the-art result but rather to provide insights into the working of multi-task learning. It is conceivable that our results can be extended to the entire dataset as well. This is left for future work. More details about the GLUE benchmark can be found in the original paper (Wang et al. (2018a)).

**ChestX-ray14.** The ChestX-ray14 dataset (Wang et al. (2017)) is the largest publicly available chest X-ray dataset. It contains 112,120 frontal-view X-ray images of 30,805 unique patients. Each image contains up to 14 different thoracic pathology labels using automatic extraction methods on radiology reports. This can be formulated as a 14-task multi-label image classification problem. The ChestX-ray14 dataset is a representative dataset in the medical imaging domain as well as in computer vision. We use this dataset to examine our proposed task reweighting scheme since it satisfies the assumption that all tasks have the same input data but different labels.

## B.2 MODELS

**Synthetic settings.** For the synthetic experiments, we use the linear regression model, the logistic regression model and a one-layer neural network with the ReLU activation function.

**Sentiment analysis.** For the sentiment analysis experiments, we consider three different models including multi-layer perceptron (MLP), LSTM, CNN:

- For the MLP model, we average the word embeddings of a sentence and feed the result into a two layer perceptron, followed by a classification layer.

- For the LSTM model, we use the standard one-layer single direction LSTM as proposed by Lei et al. (2018), followed by a classification layer.

- For the CNN model, we use the model proposed by Kim (2014) which uses one convolutional layer with multiple filters, followed by a ReLU layer, max-pooling layer, and classification layer. We follow the protocol of Kim (2014) and set the filter size as $\{3, 4, 5\}$.

We use the pre-trained GLoVe embeddings trained on Wikipedia 2014 and Gigaword 5 corpora [6]. We fine-tune the entire model in our experiments. In the multi-task learning setting, the shared modules include the embedding layer and the feature extraction layer (i.e. the MLP, LSTM, or CNN model). Each task has its separate output module.

**GLUE.** For the experiments on the GLUE benchmark, we use a state-of-the-art language model called BERT (Devlin et al. (2018)). For each task, we add a classification/regression layer on top it as our model. For all the experiments, we use the $\text{BERT}_{\text{LARGE}}$ uncased model, which is a 24 layer network as described in Devlin et al. (2018). For the multi-task learning setting, we follow the work of Liu et al. (2019a) and use $\text{BERT}_{\text{LARGE}}$ as the shared module.

**ChestX-ray14.** For the experiments on the ChestX-ray14 dataset, we use the DenseNet model proposed by Rajpurkar et al. (2017) as the shared module, which is a 121 layer network. For each task, we use a separate classification output layer. We use the pre-trained model[7] in our experiments.

## B.3 TRAINING PROCEDURES

In this subsection, we describe the training procedures for our experiments.

**Mini-batch SGD.** We describe the details of task data sampling in our SGD implementation.

- For tasks with different features such as GLUE, we first divide each task data into small batches. Then, we mix all the batches from all tasks and shuffle randomly. During every epoch, a SGD step is applied on every batch over the corresponding task. If the current batch is for task $i$, then the SGD is applied on $A_i$, and possibly $R_i$ or $B$ depending on the setup. The other parameters for other tasks are fixed.

- For tasks with the same features such as ChestX-ray14, the SGD is applied on all the tasks jointly to update all the $A_i$'s and $B$ together.

**Synthetic settings.** For the synthetic experiments, we do a grid search over the learning rate from $\{1e-4, 1e-3, 1e-2, 1e-1\}$ and the number of epochs from $\{10, 20, 30, 40, 50\}$. We pick the best results for all the experiments. We choose the learning rate to be $1e-3$, the number of epochs to be 30, and the batch size to be 50. For regression task, we report the Spearman's correlation score For classification task, we report the classification accuracy.

**Sentiment analysis.** For the sentiment analysis experiments, we randomly split the data into training, dev and test sets with percentages $80\%$, $10\%$, and $10\%$ respectively. We follow the protocol of Lei et al. (2018) to set up our model for the sentiment analysis experiments.

The default hidden dimension of the model (e.g. LSTM) is set to be 200, but we vary this parameter for the model capacity experiments. We report the accuracy score on the test set as the performance metric.

---

[6]http://nlp.stanford.edu/data/wordvecs/glove.6B.zip
[7]https://github.com/pytorch/vision

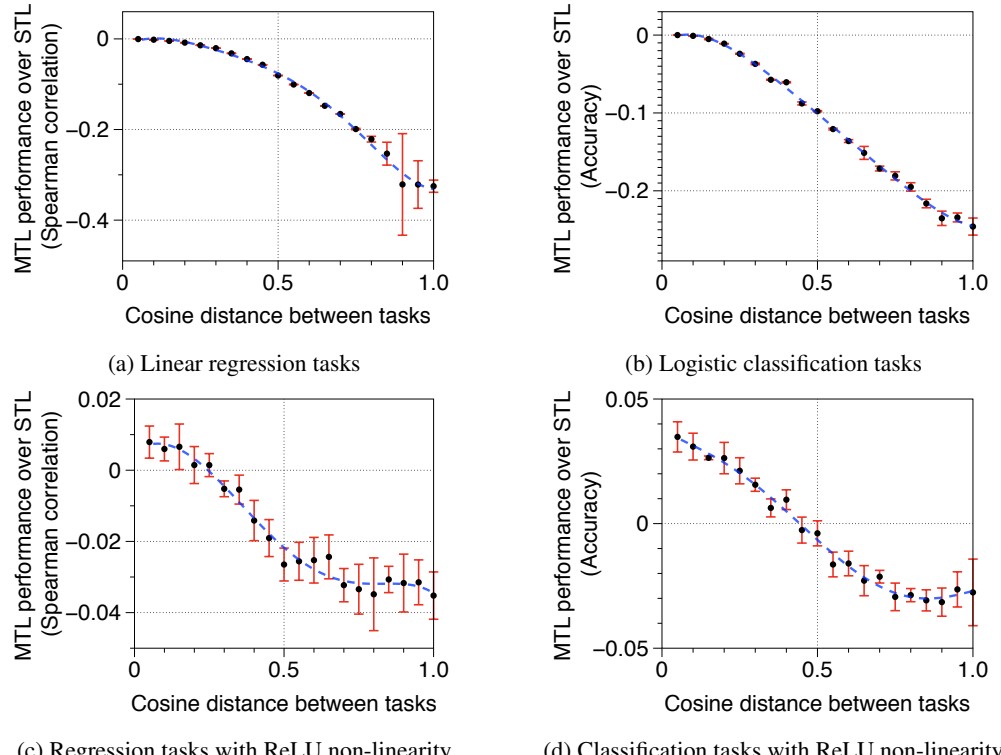

Figure 7: Comparing MTL model performance over different task similarity. For (a) and (c), MTL trains two regression tasks; For (b) and (d), MTL trains two classification tasks. For regression tasks, we use spearman correlation as model performance indicator. For classification tasks, we use accuracy as the metric. We report the average model performance over two tasks. The $x$-axis denotes the cosine distance, i.e. $1 - \cos(\theta_1, \theta_2)$.

**GLUE.** For the GLUE experiments, the training procedure is used on the alignment modules and the output modules. Due to the complexity of the $\text{BERT}_{\text{LARGE}}$ module, which involves 24 layers of non-linear transformations.

We fix the $\text{BERT}_{\text{LARGE}}$ module during the training process to examine the effect of adding the alignment modules to the training process. In general, even after fine-tuning the $\text{BERT}_{\text{LARGE}}$ module on a set of tasks, it is always possible to add our alignment modules and apply Algorithm 1.

For the training parameters, we apply grid search to tune the learning rate from $\{2e{-}5, 3e{-}5, 1e{-}5\}$ and the number of epochs from $\{2, 3, 5, 10\}$. We choose the learning rate to be $2e{-}5$, the number of epochs to be 5, and with batch size 16 for all the experiments.

We use the GLUE evaluation metric (cf. Wang et al. (2018b)) and report the scores on the development set as the performance metric.

**ChestX-ray14.** For the ChestX-ray14 experiments, we use the configuration suggested by Rajpurkar et al. (2017) and report the AUC score on the test set after fine-tuning the model for 20 epochs.

### B.4    EXTENDED SYNTHETIC EXPERIMENTS

**Varying cosine similarity on linear and ReLU models.** We demonstrate the effect of cosine similarity in synthetic settings for both regression and classification tasks.

*Synthetic tasks.* We start with linear settings. We generate 20 synthetic task datasets (either for regression tasks, or classification tasks) based on data generation procedure and vary the task similarity between task 1 and task $i$. We run the experiment with a different dataset pairs (dataset 1 and dataset $i$).

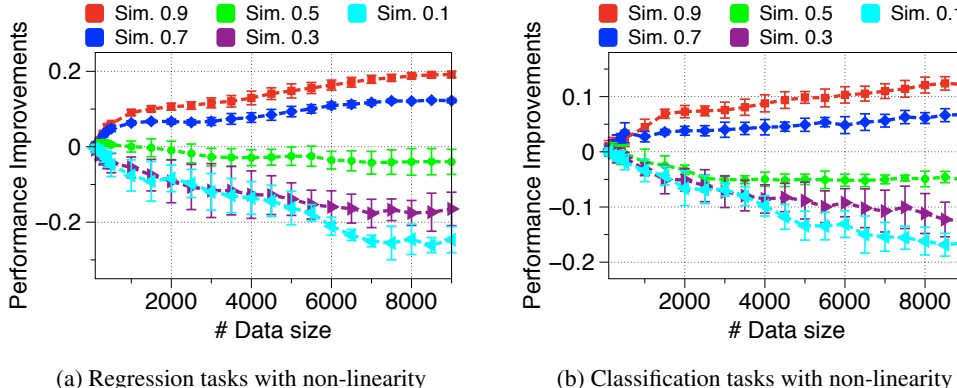

(a) Regression tasks with non-linearity      (b) Classification tasks with non-linearity

Figure 8: The performance improvement on the target task (MTL minus STL) by varying the cosine similarity of the two tasks' STL models. We observe that higher similarity between the STL models leads to better improvement on the target task.

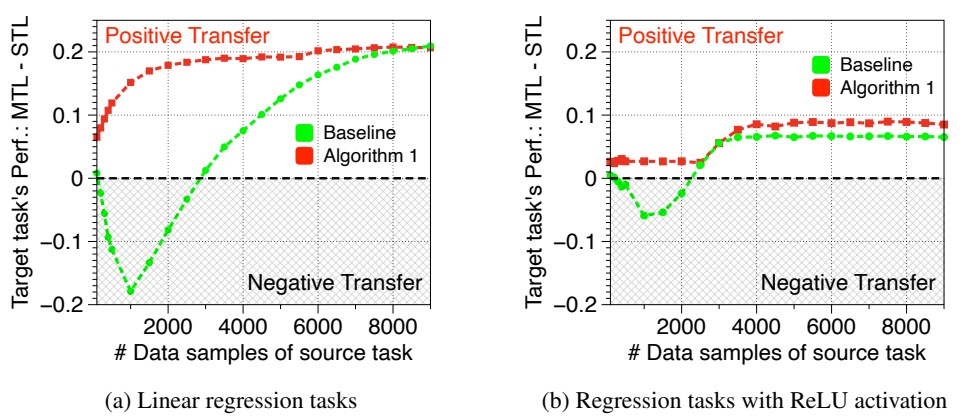

(a) Linear regression tasks      (b) Regression tasks with ReLU activation

Figure 9: Comparing Algorithm 1 to the baseline MTL training on the synthetic example in Section 2.3. Algorithm 1 corrects the negative transfer phenomenon observed in Figure 3.

After generating the tasks, we compare the performance gap between MTL and STL model.

*Results.* From Figure 7a and Figure 7a, we find that for both regression and classification settings, with the larger task similarity the MTL outperforms more than STL model and the negative transfer could occur if the task similarity is too small.

*ReLU settings.* We also consider a ReLU-activated model. We use the same setup as the linear setting, but apply a ReLU activation to generate the data. Similar results are shown in Figure 7c, 7d.

**Higher rank regimes for ReLU settings.** We provide further validation of our results on ReLU-activated models.

*Synthetic tasks.* In this synthetic experiment, there are two sets of model parameters $\Theta_1 \subseteq \mathbb{R}^{d \times r}$ and $\Theta_2 \subseteq \mathbb{R}^{d \times r}$ ($d = 100$ and $r = 10$). $\Theta_1$ is a fixed random rotation matrix and there are $m_1 = 100$ data points for task 1. Task 2's model parameter is $\Theta_2 = \alpha \Theta_1 + (1 - \alpha)\Theta'$, where $\Theta'$ is also a fixed rotation matrix that is orthogonal to $\Theta_1$. Note that $\alpha$ is the cosine value/similarity of the principal angle between $\Theta_1$ and $\Theta_2$.

We then generate $X_1 \subseteq \mathbb{R}^{m_1 \times d}$ and $X_2 \subseteq \mathbb{R}^{m_2 \times d}$ from Gaussian. For each task, the labels are $y_i = \text{ReLU}(X_i \Theta_i)e + \varepsilon_i$, where $e \in \mathbb{R}^r$ is the all ones vector and $\varepsilon_i$ is a random Gaussian noise.

Given the two tasks, we use MTL with ReLU activations and capacity $H = 10$ to co-train the two tasks. The goal is to see how different levels of $\alpha$ or similarity affects the transfer from task two to task one. Note that this setting parallels the ReLU setting of Theorem 9 but applies to rank $r = 5$.

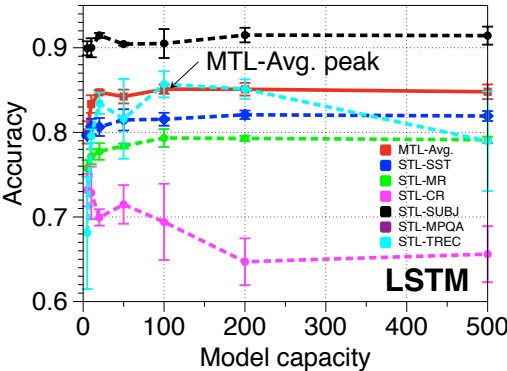

Figure 10: Cross validation to choose the best performing model capacity for each model.

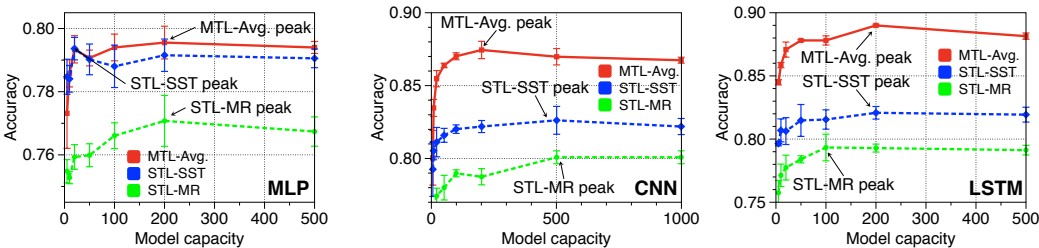

Figure 11: Validation on MLP, CNN and LSTM models for sentiment analysis tasks.

*Results.* In Figure 8 we show that the data size, the cosine similarity between the STL solutions and the alignment of covariances continue to affect the rate of transfer in the new settings. The study shows that our conceptual results are applicable to a wide range of settings.

**Evaluating Algorithm 1 on linear and ReLU-activated models.** We consider the synthetic example in Section 2.3 to compare Algorithm 1 and the baseline MTL training. Recall that in the example, when the source and target tasks have different covariance matrices, MTL causes negative transfer on the target task. Our hypothesis in this experiment is to show that Algorithm 1 can correct the misalignment and the negative transfer.

*Synthetic tasks.* We evaluate on both linear and ReLU regression tasks. The linear case follows the example in Section 2.3. For the ReLU case, the data is generated according to the previous example.

*Results.* Figure 9 confirms the hypothesis. We observe that Algorithm 1 corrects the negative transfer in the regime where the source task only has limited amount of data. Furthermore, Algorithm 1 matches the baseline MTL training when the source task has sufficiently many data points.

## B.5 EXTENDED ABLATION STUDIES

**Cross validation for choosing model capacities.** We provide a cross validation experiment to indicate how we choose the best performing model capacities in Figure 1. This is done on the six sentiment analysis tasks trained with an LSTM layer.

In Figure 10, we vary the model capacities to plot the validation accuracies of the MTL model trained with all six tasks and the STL model for each task. The result complements Table 1 in Section 3.3.

**Choosing model capacities for CNN and MLP.** Next we verify our result on model capacities for CNN and MLP models. We select the SST and MR datasets from the sentiment analysis tasks for this experiment. We train all three models CNN, MLP and LSTM by varying the capacities.

*Results.* From Figure 11 we observe that the best performing MTL model capacity is less than total best performing model capacities of STL model on all models.

**The effect of label noise on Algorithm 2.** To evaluate the robustness of Algorithm 2 in the presence of label noise, we conduct the following experiment. First, we subsample 10% of the ChestX-ray14 dataset and select two tasks from it. Then, we randomly pick one task to add 20% of noise to its labels by randomly flipping them with probability 0.5. We compare the performance of training both tasks using our reweighting scheme (Algorithm 2) vs. the reweighting techniques of Kendall et al. (2018) and the unweighted loss scheme.

*Results.* On 10 randomly chosen task pairs, our method improves over the unweighted training scheme by 1.0% AUC score and 0.4% AUC score over Kendall et al. (2018) averaged over the 10 task pairs. Figure 12 shows 5 example task pairs from our evaluation.

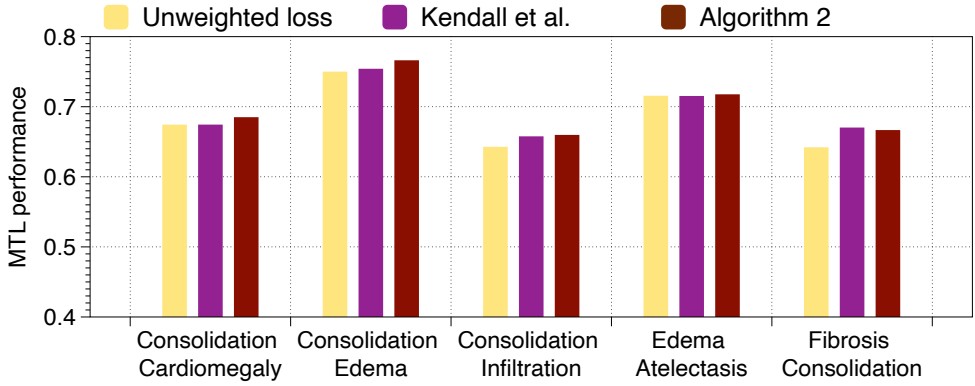

Figure 12: Comparing Algorithm 2 to the unweighted scheme and Kendall et al. (2018).

