# OpenReview forum: "Understanding and Improving Information Transfer in Multi-Task Learning"
_ICLR.cc/2020/Conference — Accept (Poster)_

### Official Review · AnonReviewer3 · 2019-10-21
**Official Blind Review #3**

**Rating:** 6

**Review:**

This paper studies how to improve the multi-task learning from both theoretical and experimental viewpoints. More specifically, they study an architecture where there is a shared model for all of the tasks and a separate module specific to each task. They show that data similarity of the tasks, measured by task covariance is an important element for the tasks to be constructive or destructive. They theoretically find a sufficient condition that guarantee one task can transfer positively to the other; i.e. a lower bound of the number of data points that one task has to have. Consequently, they propose an algorithm which is basically applying a covariance alignment method to the input.
The paper is well-written, and easy to follow.
Pros:
A new theoretical analysis for multi-task learning, which can give insight of how to improve it through data selection.
They empirically show that their algorithm improves the multi-task learning on average by 2.35%.

Cons:
There is not much of novelty in the algorithm and architecture. Their method is very similar to domain adaptation but for multi-learning setting.
In the Theorem 2, they have assumed parameter c <= 1/3. They have not provided any insight of how much restrictive this assumption is.



**Experience Assessment:**

I have read many papers in this area.

**Review Assessment: Checking Correctness Of Derivations And Theory:**

I did not assess the derivations or theory.

**Review Assessment: Checking Correctness Of Experiments:**

I assessed the sensibility of the experiments.

**Review Assessment: Thoroughness In Paper Reading:**

I read the paper thoroughly.

---

> ### Author Response · Authors · 2019-11-12
> **Response to Reviewer #3**
>
> We thank the reviewer for the positive feedback and for the appreciation of our theoretical contribution and our effort on writing. We respond to the two comments under “cons” here.
>
>   > “There is not much of novelty in the algorithm and architecture. Their method is very similar to domain adaptation but for multi-learning setting.”
>
>     - We do agree that in domain adaptation, it is well-understood that the divergence between the source and target distributions can cause negative transfer. Hence, the general recipe is to correct this divergence by matching the source distribution to the target. In the multi-task setting, however, the interaction/interference between the tasks is much more complicated, e.g. positive and negative effects can happen at the same time (e.g. figure 6). To determine the type of interference, we provide a theoretical framework to study this question in linear and ReLU models and we develop theory to identify the components which cause positive and negative transfers.
>     - We would like to emphasize that our covariance alignment algorithm and SVD-based reweighing scheme are both consequences derived from our theory. The additional experiments we added into the revision verify that the alignment algorithm can correct misaligned task data for linear models (Appendix C.5), and we have shown that it works well for highly non-linear networks (Sec. 3.2). Our insight for these empirical results is that there exists an alignment matrix that corrects the differences between the task covariances, which can cause negative effect in MTL. We believe that this insight is applicable to more sophisticated architectures.
>     - In addition to the algorithms, our theoretical framework provides some general rules of thumb and tools to help MTL in practice, including i) We show that the capacity of the shared MTL module should not exceed the total capacities of all the STL modules; ii) We propose the cosine similarity score to measure the similarities of task data and track the progress of the alignment procedure.
>
>   > “In the Theorem 2, they have assumed parameter $c <= 1/3$. They have not provided any insight of how much restrictive this assumption is.”
>
>     - In Theorem 2, the assumption that $c <= 1/3$ arises when we deal with the label noise of task 2. If there is no noise for task 2, then this assumption is not needed. If there is noise for task 2, this assumption is satisfied when $\sin(\theta_1, \theta_2)$ is less than $1/(3\kappa(X_2))$. This is satisfied when the two single-task models are close enough, which is intuitively necessary to guarantee positive transfer. Indeed our experiments also show that the value of $\sin(\theta_1, \theta_2)$ affects performance (Figure 8 and 9 in Appendix C.4).
>     - Theorem 2 guarantees positive transfers in MTL, when the source and target models are close enough and the number of source samples is large. While the intuition is folklore in MTL, we provide a formal justification in the linear and ReLU models to quantify the phenomenon.
>     - We have added these discussions to provide more insight on Theorem 2 into the revision in Appendix B.2.2.

---

### Official Review · AnonReviewer2 · 2019-10-22
**Official Blind Review #2**

**Rating:** 6

**Review:**

The submission investigates multitask learning (MTL) and develops new theories around MTL with linear models and linear+ReLU. In the experimental section, the authors improve performance in sentiment analysis on subtasks of the GLUE benchmark (building on BERT - highly non-linear neural network) and show a SVD-based task loss reweighting scheme on an multi-label image classification dataset.

The submission is overall well written though some paragraphs (2.1-2.3, in particular the example section) would benefit from additional effort towards clearer sentences. One issue with the submission is that there is a significant gap between the theory and experimental sections as theory only covers linear models and the experiments don’t include linear models and purely focus on deep networks. The benefits of a bottleneck in multitask learning are well known (based empirical results). However, it is helpful that the additional theoretical results (given strong assumptions) provide some grounding.
While the model with non-linear activation is mentioned at places, nearly all theorems rely on the linear model instead such that it might make sense to either work towards generalising the theorems or emphasising that most only apply to linear models.

Additional assumptions (1D labels, same input dimensionality across all tasks) should be emphasised to clarify limitations of all derivations. Where previous work addressed model similarity it often looks at models in the context of existing datasets (i.e. taking the data into account to describe boundaries etc) such that the emphasised novelty at looking at data similarity is to be taken with a grain of salt.

Overall, the paper contributes to the conversation around multitask learning but would benefit from comparing again external work on multitask learning (e.g. see under minor) and from bridging between theory and experiments (e.g. experiments with the models described in the theory section - linear/ReLU).

Minor:
- y is used as label and as data terminology at different parts of the text.
- the model in the first set of experiments has lower capacity than most models individually, suggesting that the capacity should be smaller even for individual tasks to prevent overfitting.
- An ablation over model capacities is mentioned but missing for 3.3
- comparison against existing multitask loss weighting techniques should be performed [1]


[1] Multi-Task Learning Using Uncertainty to Weigh Losses for Scene Geometry and Semantics
Alex Kendall, Yarin Gal, Roberto Cipolla 2017


**Experience Assessment:**

I have published one or two papers in this area.

**Review Assessment: Checking Correctness Of Derivations And Theory:**

I assessed the sensibility of the derivations and theory.

**Review Assessment: Checking Correctness Of Experiments:**

I carefully checked the experiments.

**Review Assessment: Thoroughness In Paper Reading:**

I read the paper at least twice and used my best judgement in assessing the paper.

---

> ### Author Response · Authors · 2019-11-12
> **Response to Reviewer #2**
>
> We thank the reviewer for appreciating the contribution of our work, and we are grateful to the suggestions which help improve our work. We added experiments to evaluate our method on linear and ReLU models regarding the suggested gap between our theory and the experiments. We added the related work [1] and provided comparative experiments with [1]. We addressed the parts that are confusing as suggested. Here are our responses regarding each comment.
>
> Response to major comments:
>
>   > “One issue with the submission is that there is a significant gap between the theory and experimental sections as theory only covers linear models and the experiments don’t include linear models and purely focus on deep networks.”
>
>     - We have added experiments to evaluate our alignment method (Alg 1) on linear and ReLU-activated models for synthetic data (Appendix C.4). Our method can indeed help these models in addition to deep networks.
>
>   > “Additional assumptions (1D labels, same input dimensionality across all tasks) should be emphasised to clarify limitations of all derivations.”
>
>     - We have revised Sec. 2.1 to emphasize that the labels are 1D in our model (the same input dimensionality assumption is also stated in Sec. 2.1). A multi-label problem with k types of labels can be modeled by k tasks with the same covariates but different labels.
>
>   > “Where previous work addressed model similarity it often looks at models in the context of existing datasets (i.e. taking the data into account to describe boundaries etc) such that the emphasised novelty at looking at data similarity is to be taken with a grain of salt.”
>
>     - We agree that if we compare two task models trained from the datasets, their similarity already depends on the data. In our experience in looking at LSTM/CNN/MLP models on sentiment analysis, however, we have observed that just measuring the similarity of the model weights or feature outputs is too crude to tell whether or not MTL shows positive or negative benefit, even for a single layer. This is likely due to the differences between the task data and the specific model used. And there is currently no theoretical framework to answer this question.
>     - To provide a more precise answer, we formulate this question in a simple setup. Our theory disentangles the model part and the data part. By doing so, figure 2 shows that task data similarity plays a second-order effect after controlling model similarity to be the same. Our intuition is that this arises from the shift of the covariance matrices between the task data. Our theory formalizes the covariances and our experiments show the benefit of aligning the covariance matrices on deep networks.
>     - We have revised the third paragraph in the intro to make it more clear.
>
>   > “While the model with non-linear activation is mentioned at places, nearly all theorems rely on the linear model instead such that it might make sense to either work towards generalising the theorems or emphasising that most only apply to linear models.”
>
>     - We thank Reviewer 2 for pointing these out. We have extended the theoretical result of Sec. 2.2 so that it applies to ReLU settings. The theoretical result of section 2.3 also applies to ReLU settings. So the only result which does not apply to the ReLU setting is proposition 3 in Sec. 2.4. The question of characterizing the optimization landscape in non-linear ReLU models is not well-understood based on the current theoretical understanding of neural networks. We think this is an open research direction and we have stated this question in the revised version.
>
> Response to minor comments:
>
>   > “y is used as label and as data terminology at different parts of the text”
>
>     - Thanks for pointing out this issue. We have corrected the use of y in the revision.
>
>   > “the model in the first set of experiments has lower capacity than most models individually, suggesting that the capacity should be smaller even for individual tasks to prevent overfitting. An ablation over model capacities is mentioned but missing for 3.3”
>
>     - We have added an ablation study over model capacities to show the performance of MTL and STL as we vary the capacities (Appendix C.5). This indicates the best performing capacities we choose in Figure 6. We also added plots on CNN/MLP to show the same results.
>
>   > “comparison against existing multitask loss weighting techniques should be performed [1]”
>
>     - Thanks for pointing out this work. We have compared our method (Alg 1) to the techniques in [1]. This is added as a benchmark in Sec. 3.1. Our SVD-based scheme performs favorably on the ChestX-ray14 dataset. The results are in Sec. 3.2 and ablation results are in Sec. 3.3 and Appendix C.5. Across all 14 tasks, our scheme outperforms [1] by 1.3% AUC score.

---

### Official Review · AnonReviewer1 · 2019-10-22
**Official Blind Review #1**

**Rating:** 8

**Review:**

This paper analyzed the principles for a successful transfer in the hard-parameter sharing multitask learning model. They analyzed three key factors of multi-task learning on linear model and relu linear model: model capacity (output dimension after common transformation), task covariance (similarity between tasks) and optimization strategy (influence of re-weighting algorithm), with theoretical guarantees. Finally they evaluated their assumptions on the state-of-the-art multi-task framework (e.g GLUE,CheXNet), showing the benefits of the proposed algorithm.

Main comments:

This paper is highly interesting and strong. The author systematically analyzed the factors to ensure a good multi-task learning. The discovering is coherent with with previous works, and it also brings new theoretical insights (e.g. sufficient conditions to induce a positive transfer in Theorem 2). The proof is non-trivial and seems technically sound.

Moreover, they validated their theoretical assumptions on the large scale and diverse datasets (e.g NLP tasks, medical tasks) with state-of-the-art baselines, which verified the correctness of the theory and indicated strong practical implications.

Minor comments:
The main message of the paper is clear but some parts still confuse me:
1. I suggest the author to merge the Figure 3 and Data generation (Page 4) part for a better presentation. e,g which “diff.covariance” is task 3 or 4 ?  And why we use different rotation matrix Q_i ?

2. In algorithm 1 (Page 5) , I suggest the author use a formal equation (like algorithm 2) instead of descriptive words.
     -- Step 2, I have trouble in understading this step.
     -- Step 3, how to jointly minimize R_1,\dots, R_k, A_1, \dots, A_k ? we use loss (3)  or other losses ?
     -- I suggest that the author release the code for a better understanding.

3. For theorem 2, can we find some “optimal” c to optimize the right part ? Since 6c + \frac{1}{1-3c}\frac{\epsilon}{\X_2\theta_2} might be further optimized

4. In section 3.3. (Figure 6)  of the real neural network, the model capacity is the dimension of Z or simply the dimension before last fc-layer ?

5. Some parts in the appendix can be better illustrated:
    (a) I am not clear how proposition 4 can derive proposition 1.
    (b) Page 15, proving fact 8: last line \frac{1}{k^4}sin(a^{prime},b^{prime}) should be \frac{1}{k^4}sin^{2}(a^{prime},b^{prime}).


Overall I think it is a good work with interesting discoverings for the multi-task learning. I think it will potentially inspire the community to have more thoughts about the transfer learning.


**Experience Assessment:**

I have published one or two papers in this area.

**Review Assessment: Checking Correctness Of Derivations And Theory:**

I assessed the sensibility of the derivations and theory.

**Review Assessment: Checking Correctness Of Experiments:**

I assessed the sensibility of the experiments.

**Review Assessment: Thoroughness In Paper Reading:**

I read the paper at least twice and used my best judgement in assessing the paper.

---

> ### Author Response · Authors · 2019-11-12
> **Response to Reviewer #1**
>
> We thank the reviewer for appreciating our theoretical insights and the strength of our work. We appreciate the reviewer’s effort to provide detailed comments which we have incorporated in the revision. Here are our changes with respect to each comment.
>
>   > “1. I suggest the author to merge the Figure 3 and Data generation (Page 4) part for a better presentation. e,g which “diff.covariance” is task 3 or 4?  And why we use different rotation matrix Q_i ?”
>
>     - We have revised Figure 3 and the Data generation paragraph in Sec. 2.3 to clarify the two issues.
>     - Figure 3 is simplified to 2 curves without affecting the message. The caption connects the figure to the generation process. And the generation process is revised accordingly in reference to the figure.
>     - The different rotation matrices Q_i are used to create a “covariate shift” between the two tasks. As Figure 3 shows, this shift leads to a negative transfer of MTL in the regime where the number of source data points is small.
>
>   > “2. In algorithm 1 (Page 5), I suggest the author use a formal equation (like algorithm 2) instead of descriptive words.
>      -- Step 2, I have trouble in understanding this step.
>      -- Step 3, how to jointly minimize R_1,\dots, R_k, A_1, \dots, A_k ? we use loss (3)  or other losses?
>      -- I suggest that the author release the code for a better understanding.”
>
>     - We have revised the description of algorithm 1 to define a formal loss. To minimize the modified loss over the alignment matrices and the output layers, we use standard training procedures (mini-batch SGD, cf. Appendix C.3). Our code will be open-sourced after the reviewing process.
>
>   > “3. For theorem 2, can we find some “optimal” c to optimize the right part ? Since 6c + \frac{1}{1-3c}\frac{\epsilon}{\X_2\theta_2} might be further optimized.”
>
>     - The error bound $6c + \frac{1}{1-3c}\frac{||\epsilon||}{||X_2\theta_2||}$ decreases with c so the smaller c is the better. We have revised Theorem 2 and added a discussion regarding the error bound (Appendix B.2.2).
>
>   > “4. In section 3.3. (Figure 6)  of the real neural network, the model capacity is the dimension of Z or simply the dimension before last fc-layer?”
>
>     - The model capacity is the dimension before the last fc-layer. We have revised this sentence to make it clear.
>
>   > “5. Some parts in the appendix can be better illustrated:
>     (a) I am not clear how proposition 4 can derive proposition 1.
>     (b) Page 15, proving fact 8: last line \frac{1}{k^4}sin(a^{prime},b^{prime}) should be \frac{1}{k^4}sin^{2}(a^{prime},b^{prime}).”
>
>     - (a) We have revised proposition 4 so that it becomes more clear that it can derive proposition 1. In particular, proposition 4 states that the subspace of the shared module is all that matters, hence having the $\{\theta_i\}$’s in its column span suffices. Proposition 1 instantiates this intuition.
>     - (b) Thanks for catching the typo. We have fixed it.

---

> > ### Comment · AnonReviewer1 · 2019-11-14
> > **Response**
> >
> > Thanks for your detailed rebuttal and efforts for making the paper better.
> >
> > Most of my confusions have been solved and the only unclear point might be in question (2):
> >
> > “Step 3, how to jointly minimize R_1,\dots, R_k, A_1, \dots, A_k ?”
> > I am wondering the high level solution. For example If we are only considering a simple linear model, jointly optimization means that we simply set a global parameter $\phi  = (R_1,\dots, R_k, A_1, \dots, A_k) $ and apply gradient descent over $\phi$ of the loss (seems more difficult).  Or we use alternative optimization (seems more common),  for each optimization step we fix all parameters except one parameter and optimize only that one parameter.
> >
> > Overall I keep my current decision and think it is indeed a good paper.

---

> > > ### Author Response · Authors · 2019-11-15
> > > **Thanks for raising this question**
> > >
> > > We use alternative optimization in our implementation of Alg 1. For each epoch, we iterate over all the task batches. If the current batch is from task $i$, then the SGD is applied on $A_i$ and $R_i$. The other parameters are fixed. We have revised Alg 1 to clarify this step and also included the description of our SGD implementation in Appendix C.3.

---

> > ### Public Comment · ~Joshua_Yee_Kim1 · 2020-09-27
> > **Open Source Code**
> >
> > Hi Authors,
> >
> > Thank you for the very interesting work. Could I please follow-up on link to the open source code regarding, "Our code will be open-sourced after the reviewing process."
> >
> > Best Regards,
> > Joshua

---

> > > ### Author Response · Authors · 2023-08-01
> > > **Re: Open Source Code**
> > >
> > > Dear Joshua,
> > >
> > > Thanks for your inquiry. We built all of our multitask learning experiments using emmental, which can be publicly accessed in the following GitHub repository:
> > >
> > > https://github.com/senwu/emmental
> > >
> > > We also have tutorials using emmental to build sentiment analysis and GLUE experiments used in our paper. They were open-sourced a long time ago and can be accessed in the following GitHub repository:
> > >
> > > https://github.com/SenWu/emmental-tutorials
> > >
> > > If you need help with setting up experiments in our paper, feel free to email us. We're happy to help you set things up.
> > >
> > > Apologies for the delay in response.
> > >
> > > Hongyang

---

### Author Response · Authors · 2019-11-12
**Summary of the revision and the response**

We thank all the reviewers for the positive feedback and the detailed comments. In response to the reviewers’ suggestions, we have revised our paper, including three sets of additional experimental results to consolidate our results as follows.

  1- Clarify our model assumption on 1D label and how to model multi-label problems, the example of figure 3 and the data generation description, and the theory part (formal description of Alg 1, extendable to ReLU or not, discussion on Theorem 2 in Appendix B.2.2).

  2- Additional experiments on linear and ReLU models to validate our alignment method (Appendix C.5). This confirms that our method (Alg 1) can help linear models in addition to deep networks, as Reviewer #2 asked about.

  3- We conduct an additional experiment to compare our SVD-based reweighting scheme to the loss weighting techniques of Kendall et al.’18, as Reviewer #2 requested. On the ChestX-ray14 dataset, we found that our method improves performance by 1.3% AUC score compared to the suggested work (Sec. 3.2).

  4- Additional ablation studies on model capacities to further validate our results (Appendix C.5), as Reviewer #2 asked about.

Finally, we respond to all the comments raised by the reviewers in detail. The comments have all been incorporated into the revision.

---

### Decision · Program_Chairs · 2019-12-19

**Decision:**

Accept (Poster)

**Comment:**

Many existing approaches in multi-task learning rely on intuitions about how to transfer information. This paper, instead, tries to answer what does "information transfer" even mean in this context. Such ideas have already been presented in the past, but the approach taken here is novel, rigorous and well-explained.

The reviewers agreed that this is a good paper, although they wished to see the analysis conducted using more practical models.

For the camera ready version it would help to make the paper look less dense.